# Light-driven biological actuators to probe the rheology of 3D microtissues

Adrien Méry [1], Artur Ruppel[1], Jean Revilloud[1], Martial Balland [1], Giovanni Cappello [1] & Thomas Boudou[1] ✉

The mechanical properties of biological tissues are key to their physical integrity and function. Although external loading or biochemical treatments allow the estimation of these properties globally, it remains difficult to assess how such external stimuli compare with cell-generated contractions. Here we engineer microtissues composed of optogenetically-modified fibroblasts encapsulated within collagen. Using light to control the activity of RhoA, a major regulator of cellular contractility, we induce local contractions within microtissues, while monitoring microtissue stress and strain. We investigate the regulation of these local contractions and their spatio-temporal distribution. We demonstrate the potential of our technique for quantifying tissue elasticity and strain propagation, before examining the possibility of using light to create and map local anisotropies in mechanically heterogeneous microtissues. Altogether, our results open an avenue to guide the formation of tissues while non-destructively charting their rheology in real time, using their own constituting cells as internal actuators.

Tissue engineering holds great potential to develop organotypic in vitro model systems. With their miniaturization in the late 2000's[1,2], such three-dimensional (3D) microtissue models have been used for studying fundamental features of tissue biology such as the repair of wounded fibrous tissue[3], or the formation and maturation of myocardial[4] and lung tissues[5]. Their sub-millimetric size has also opened the possibility to use high-throughput, low volume screening of drugs[6,7] or functional effects of patient-specific mutations[8].

By combining microtissue engineering with magnetic or vacuum actuation, the tissue mechanical properties have recently been assessed and their importance in the tissue formation and physiological function evidenced[9,10]. For example, Reich et al. demonstrated the respective roles of cell-generated forces and collagen structures in tissue mechanics[9,11,12] while Pelling et al. connected remodeling of the cytoskeleton to homeostatic mechanical regulation of tissues[10,13]. However, using magnetic or vacuum actuations of cantilevers is restricted, by the fixed size of the cantilever, to the characterization of the global mechanical properties of microtissues, and rheological measurements have been shown to be very sensitive to the probe size and the loading history of the mechanical test device used for such characterization[14,15]. Moreover, such externally applied forces are hardly comparable to cell-generated forces, which constrains their use to study the spatio-temporal regulation of cell forces in tissues. Biochemical treatments, on the other hand, allow the modulation of the signalization pathways responsible for these processes but their inability to target spatially-defined areas of tissues and their low temporal resolution severely limit their potential for further understanding how cell forces are generated, propagated and sensed in physiological and pathological tissues. Indeed, cells are constantly pulling and pushing on their microenvironment, thus assessing, among other things, its mechanical properties[16], and probing the mechanical properties of the tissue "from the inside", as the cells naturally do it, would further our understanding of how cells investigate their environment and how cell-generated mechanical signals propagate. Yet, no method currently exists that can locally modulate the contractility of a group of cells within a 3D tissue while simultaneously measuring global tissue contractility as well as both fine-scale cytoskeletal and extracellular architecture.

Thanks to its spatial and temporal resolution, optogenetics has emerged as a powerful tool for spatiotemporally controlling cell

[1]Laboratory of Interdisciplinary Physics (LIPhy), University Grenoble Alpes, CNRS, F-38000 Grenoble, France. ✉e-mail: thomas.boudou@cnrs.fr

signaling[17]. Using optogenetic control of RhoA activity, several studies recently used light pulses to locally up- or down-regulate cell-generated forces[18–20]. Valon et al. further demonstrated that these changes in cellular tension were paralleled by tissue deformations in 2D epithelial monolayers[19], suggesting an exciting avenue for using cells as mechanical actuators. Such biological actuators could be employed to probe tissue mechanics using light-induced physiological mechanical stimuli.

We combine here 3D microtissues and optogenetics, by engineering 3D microscale constructs of optogenetically-modified fibroblasts embedded within collagen 3D matrices. Using light to control the proximity of RhoA and one of its activators ARHGEF11, we modulate cellular contractility within 3D fibrous microtissues, while microcantilevers report microtissue stress in real time. We demonstrate our ability to control and measure, over space and time, the stress of specific parts of microtissues while simultaneously inferring tissue strain using particle image velocimetry (PIV). We thus investigate the rheology of microtissues, before demonstrating the potential of our technique for quantifying the impact of the rigidity of the cantilever, the extracellular matrix and the differentiation of fibroblasts into myofibroblasts on the tissue elasticity. We then demonstrate the ability of our approach to map local anisotropies in mechanically heterogeneous microtissues, as well as to influence tissue architecture using repetitive stimulations during tissue formation. Together, these results highlight a unique approach to examine the effects of various parameters such as mechanical preload, tissue maturation, matrix architecture and stiffness on both the ability to propagate physiological mechanical signals and the dynamic mechanical properties of engineered fibrous microtissues.

## Results

### Optogenetic stimulation of microtissues

We used optogenetics to generate contractions in parts of engineered microtissues (Fig. 1a, b). Our strategy was to trigger the activation of the small GTPase RhoA, a major regulator of cellular contraction[21]. To this end, we used NIH 3T3 cells stably expressing a Cry2-CIBN optogenetic probe (opto-RhoA fibroblasts) to dynamically control with blue light the localization of ArhGEF11, an upstream regulator of RhoA[22]. As previously described, ArhGEF11 was recruited to the cell membrane upon blue light stimulation, thus activating RhoA and subsequently cell contractility (Fig. 1c)[19,20]. We embedded these opto-RhoA fibroblasts in a neutralized collagen I solution within microfabricated PDMS wells containing two T-shaped microcantilevers. Over time of cultivation, the fibroblasts spread inside the collagen and spontaneously compacted the matrix to form opto-RhoA microtissues that spanned across the top of the pair of cantilevers (Fig. 1a), which deflection was measured to quantify tissue tension[2]. During tissue formation, a baseline static tension developed due to the compaction of the gel by the collective action of the fibroblasts. Once the tissue formed, we used a digital micro-mirror device (DMD) to illuminate parts of these opto-RhoA microtissues. To demonstrate our ability to apply local stimulations, we consecutively illuminated the left and right halves of microtissues while measuring the microtissue tension as well as the local displacements using PIV (Fig. 1d–f, Supplementary movie 1). Immediately after a pulse of light (characterized by an irradiance of 1.8 mW/mm² and a duration of 500 ms, i.e. an energy of 0.9 mJ/mm²), the tissue in the stimulated area contracted, inducing a quick increase in tissue tension (half-time to maximum increase $t_{\frac{1}{2}}^{i} = 26 \pm 8$ s), before slowly relaxing back to baseline (half-time of decrease from maximum $t_{\frac{1}{2}}^{d} = 317 \pm 54$ s) (Fig. 1d–f). Figure 1f shows the tension increase corresponding to the local contractions of the left and right halves of the microtissues. For both stimulations patterns, the microtissue tension increased to a similar level and decreased back to baseline level, thus indicating an elastic response without plastic deformation. By engineering microtissues composed of wild type (WT)

NIH 3T3 fibroblasts, we confirmed these light-induced contractions were due to the optogenetic construction and inexistent in WT microtissues (Fig. 1d–f). Of note, opto-RhoA microtissues were broader and generated more tension than WT microtissues, but both generated similar baseline stress $\sigma_{xx}$ (i.e. tissue tension obtained from the cantilever deflection divided by the cross-sectional area in the center of the tissue) (Fig. 1g). Furthermore, the local displacements were mostly directed along the longitudinal axis, suggesting a strongly anisotropic contraction. To further investigate the geometry of deformation, we next stimulated a discoidal area of microtissues suspended between two and four cantilevers.

### Light-induced contractions depend on the microtissue architecture

A major advantage of optogenetics over drug or genetic approaches is the possibility to stimulate spatially defined areas of a tissue of interest. We thus investigated the mechanical response of opto-microtissues of which only a small area is stimulated with blue light. We stimulated a 50 μm diameter disc in the center of the microtissue (Fig. 2a, Supplementary movie 2) and mapped the resulting displacements over the whole microtissue by using particle image velocimetry (PIV). We observed that the stimulated central region barely moved upon illumination while the left and right parts of the microtissues were displaced inwards, toward each other, indicating compaction of the stimulated area (Fig. 2b). Despite a discoidal light stimulation, the resulting displacements were strongly polarized, with a mean angle of $16 \pm 7°$ and more than 80% of the displacements presenting a lower than 30° angle with the longitudinal x-axis of the tissue (Fig. 2c). This polarized displacement field corresponds to a strongly anisotropic field of deformation, as the x-component of the strain ($\varepsilon_{xx}$) was more than 15-times larger than its y-component ($\varepsilon_{yy}$) (Fig. 2d–f, Supplementary movie 2). We defined the anisotropy coefficient (AC) as the difference between $\varepsilon_{xx}$ and $\varepsilon_{yy}$, normalized by their sum. In the case of these microtissues suspended between two cantilevers, we measured an AC of $1.2 \pm 0.2$, indicating a strong anisotropy along the x-axis. This quasi-1D response suggests a strong anisotropy of either the cytoskeleton of the constituting cells or the surrounding matrix. We simultaneously stained actin and collagen in microtissues and quantified their alignment. The confocal images showed a compacted collagen core, sparsely populated with fibroblasts, surrounded by a highly cellularized peripheral shell (Supplementary movie 3), consistent with previous observations[9,23]. We also found that the longitudinal contraction induced by light correlated with a strongly anisotropic architecture of both actin and collagen, with more than 80 % of the actin fibers and more than 70 % of the collagen fibers aligned within less than 30° of the x-axis (Fig. 2g–l, Supplementary movie 3). As cortical actin is prominent, cell alignment correlates with actin alignment, which is coherent with the fact that fibroblasts line up and remodel their extracellular matrix to align with the principal maximum strains developed during tissue formation[24–26]. Furthermore, we show here using optogenetics that any additional contraction, even local, is constrained by the anisotropic conformation of the tissue.

### Dynamic of optogenetically-induced strains

Having shown the quasi-1D deformation of anisotropic microtissues spanning between two cantilevers, we next sought to further investigate the spatio-temporal strain patterns induced by such local, optogenetically-induced contractions. We stimulated the left half of microtissues and used PIV to map the resulting displacements and infer the corresponding strain $\varepsilon_{xx}$ (Fig. 3, Supplementary movie 4). We observed that the stimulated half left of the microtissue was compressed, up to $\varepsilon_C^{max} = -3.0 \pm 0.5$ %, while the non-stimulated half right of the tissue was stretched, up to $\varepsilon_S^{max} = 1.0 \pm 0.3$ %. This result indicates that when stimulated cells contract, compact and compress their surroundings. This compression is equilibrated by both the stretch of

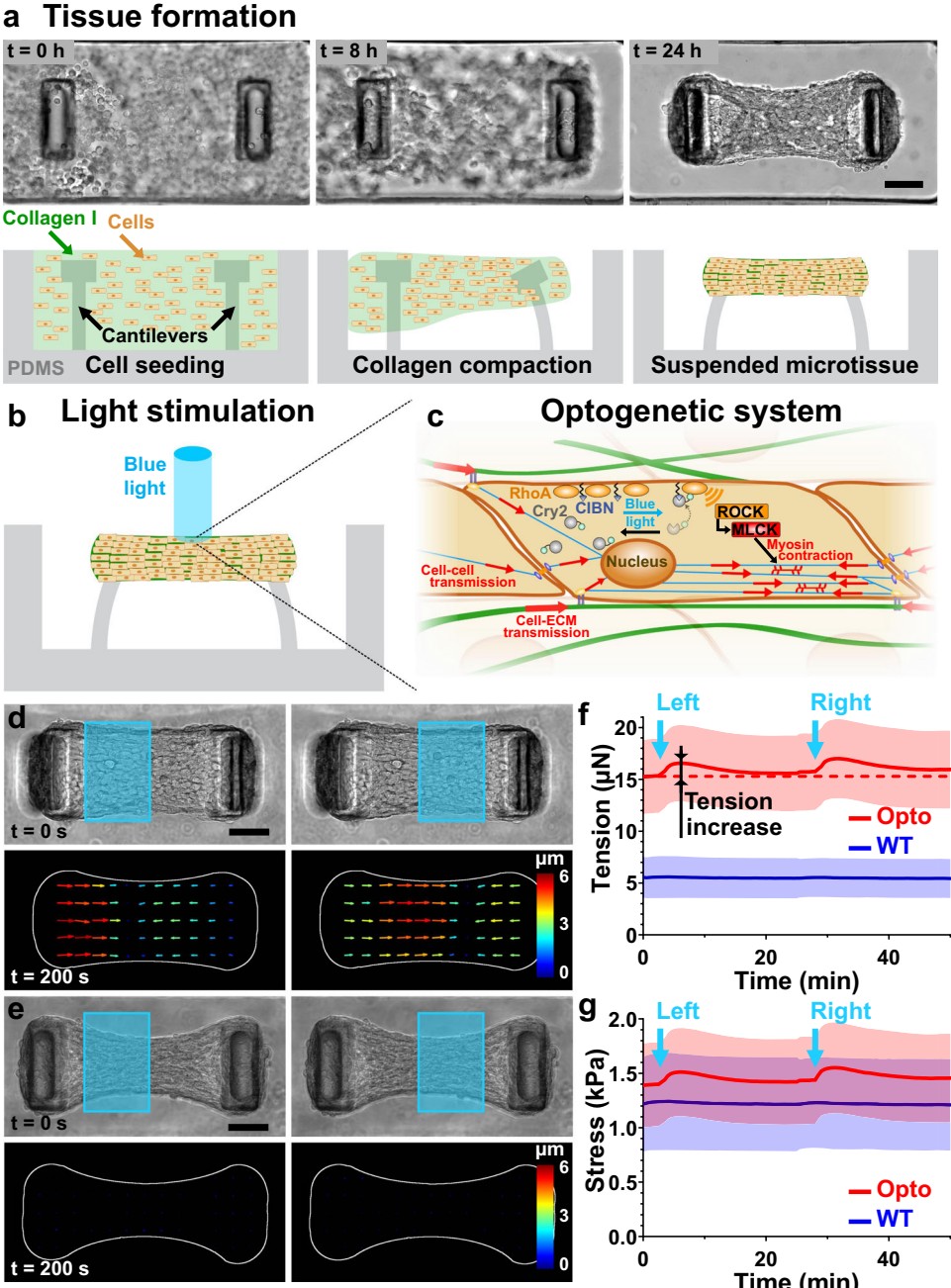

**Fig. 1 | Light-controlled contraction of microtissues. a** Representative top view images, among 21 microtissues over 3 independent experiments, and side view schemes depicting the formation of a microtissue. Over the first 24 h of cultivation, the fibroblasts spread and spontaneously compact the collagen matrix. The two T-shaped cantilevers anchor and constrain the contraction of the collagen matrix to form a microtissue that spans across the top of the pair of cantilevers. **b** Schematic illustrating the light-stimulation of a microtissue composed of Opto-RhoA fibroblasts. **c** Schematic of the ARHGEF11(DHPH)-CRY2/CIBN-CAAX optogenetic system to control cellular contractility. Upon blue light stimulation, CRY2 changes conformation and binds to CIBN, bringing the RhoA-activator ARHGEF11 in close proximity of RhoA, which in turn induces a contractility increase. **d** Upon blue light illumination of its left- or right-half, an opto-RhoA microtissue contracts locally, as shown by the PIV-tracking of the displacements, whereas a WT microtissue is unaffected (**e**). For readability reasons, only half of the vectors are represented. Scale bar is 100 μm. Temporal evolution of the tension (**f**) and stress $\sigma_{xx}$ (**g**) generated by Opto-RhoA (in red) or WT (in blue) microtissues upon the simulation of their left- and right-half. Data are the average of $n = 20$ microtissues over 3 independent experiments ±SD. Source data are provided as a Source Data file.

the non-stimulated part of the tissue and the deflection of the cantilevers. Figure 3c, d shows the evolution of tissue stress $\sigma_{xx}$, compression and stretch over time, indicating two time delays $\tau_C = 70 \pm 39\,s$ between the maximum stress and the maximum compression, and $\tau_S = 200 \pm 64\,s$ between the maximum stress and the maximum stretch.

By stimulating only one part of the microtissue, we were thus able to simultaneously measure both the stress $\sigma_{xx}$ and the strain $\varepsilon_{xx}$, inferred from the PIV tracking of local displacements (Fig. 3e). These results suggest the possibility to use optogenetic stimulation for assessing the mechanical properties of the microtissue.

**Inferring mechanical properties of stretched areas**
In order to probe the mechanical properties of the microtissue, we used the stimulated half of the tissue as a mechanical actuator stretching the non-stimulated half. We first compared the amplitudes of stress and stretch while varying the width of the stimulated area

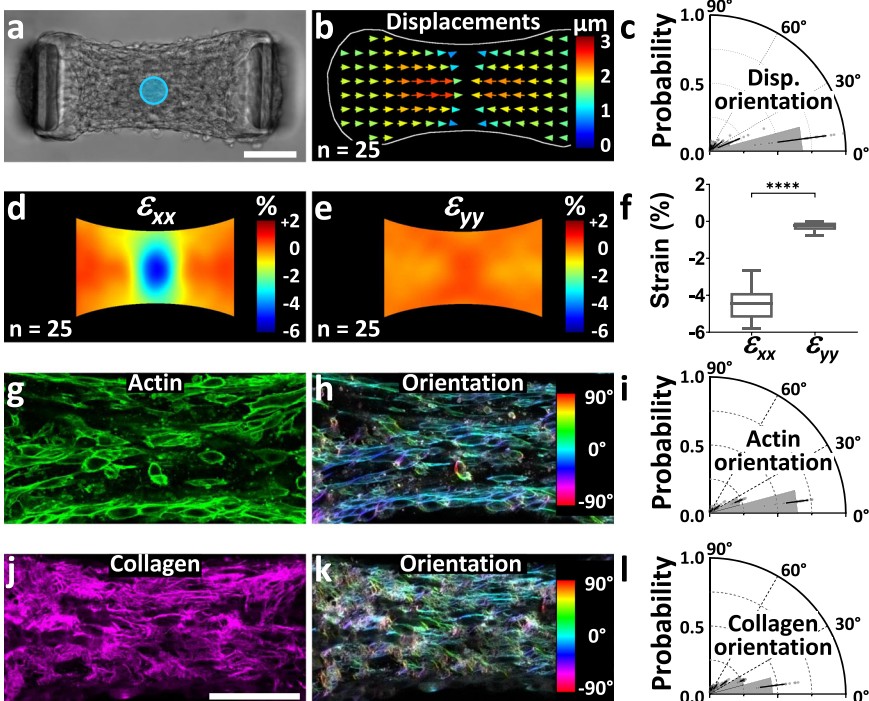

**Fig. 2 | The anisotropy of the optogenetically-induced contraction correlates with the microtissue architecture. a** Representative microtissue where the blue disc represents the 50 μm diameter isotropic stimulation area. **b** Resulting average displacement field (for readability reasons, only half of the vectors are represented) and **c** polar plot of its orientation. Data are presented as mean ± SD with $n = 25$ microtissues over 2 independent experiments superimposed with a dot plot of the data distribution. Resulting average strain fields ($\varepsilon_{xx}$ in **d**, $\varepsilon_{yy}$ in **e**) and comparison of the average strain amplitudes in the area of stimulation (**f**). Data are presented as Tukey box plots (i.e. the box extends from the 25th to 75th percentiles, the median is plotted as a line inside the box and the whiskers extend to the most extreme data point that is no more than 1.5 times the interquartile range from the edge of the box) with $n = 25$ microtissues over 2 independent experiments. ****$P < 0.0001$ determined by a two-tailed $t$-test between $\varepsilon_{xx}$ and $\varepsilon_{yy}$. **g** Confocal slice of the fluorescent staining of actin in a representative microtissue and **h** corresponding color-coded map of the actin orientation. **i** Polar plot showing the anisotropic orientation of actin fibers along the x-axis. **j** Confocal slice of the fluorescent staining of collagen in a representative microtissue and **k** corresponding color-coded map of the collagen orientation. **l** Polar plot showing the anisotropic orientation of collagen fibers along the x-axis. Data in (**i**) and (**l**) are presented as mean ± SD with $n = 27$ microtissues over 4 independent experiments superimposed with a dot plot of the data distribution. Scale bars are 100 μm. Source data are provided as a Source Data file.

from 20 to 100 μm (Fig. 4, Supplementary movie 5). We thus demonstrated that the force increase was proportional to the area of the stimulated zones (Fig. 4a and Supplementary Fig. 1a, b). As the cell density is roughly homogenous along the whole length of the microtissue (Supplementary Fig. 1c–h), this result indicates that the light-induced contraction is directly proportional to the number of stimulated cells. As the contraction is quasi-1D, the cross-section of the microtissue is unchanged when the width of stimulation is varied, and the resulting, optogenetically-induced stress increase is also proportional to the width of stimulation, ranging from $98.5 \pm 11.2$ Pa to $223.9 \pm 32.7$ Pa when the stimulation width increases from 20 to 100 μm, respectively (Fig. 4c).

As a result, the average stretch of the non-stimulated part increased from $0.6 \pm 0.2$ % to $1.2 \pm 0.3$ % for the same range of stimulation width (Fig. 4b, c, Supplementary movie 5). Upon actuation by the stimulated part, the non-stimulated part of the tissue thus exhibits viscoelastic characteristics when undergoing deformation. The maximum stretch $\varepsilon_S^{max}$ is linearly depending on the maximum applied stress $\sigma_{max}$, although peaking with a delay $\tau_S$, and both stress and $\varepsilon_{xx}$ strain returns to 0 after stimulation. Consequently, we defined the apparent elastic modulus of the tissue $E$ as the maximum stress $\sigma^{max}$ divided by the maximum stretch $\varepsilon_S^{max}$ and observed that $E$ was logically independent of the stimulation width, with an average value of $19.2 \pm 5.0$ kPa (Fig. 4d). We did not measure any significant difference in the time delay $\tau_S$ between maximum stress and stretch when varying the size of the stretched part (Fig. 4d), thus demonstrating that this delay is not a poroelastic effect occurring through the redistribution of

the fluid within the microtissue but rather to the relaxation of the actomyosin cytoskeleton.

Of note, although the range of possible frequencies is limited compared to external actuators, the stress-strain curves we obtained could be analyzed with rheological models commonly used for collagenous microtissues (Supplementary Fig. 2), such as the Kelvin-Voigt model (a spring and a dashpot in parallel), the standard linear solid (SLS) model (a spring and a dashpot in series, in parallel to another spring), or the stretched exponential (SE) model (a sum of Maxwell bodies, i.e. a spring and a dashpot in series, with a specific distribution of time constants)[13,27]. Fitting such models to the data presented in Fig. 4 for a stimulation width of 50 μm lead to consistent relaxation times τ between 101 and 135 s for the three different models. Although the Kelvin Voigt model captured the overall shape of the mechanical response and gave an elastic modulus of 18.7 kPa close to our experimental measurements, it failed to capture both the short- and long-time behavior. The SLS and the SE models lead to similar elastic constants ($E_1 = 56.3$ kPa and $E_2 = 14.7$ kPa for the SLS model, $E_1 = 65.0$ kPa and $E_2 = 10.0$ kPa for the SE model), but the SE model allowed for better fitting the long-time recovery response, thanks to the dimensionless constant $\beta = 0.66$ that captures a specific distribution of timescales (i.e. when $\beta = 1$, the SE model behaves as a SLS model, whereas when β decreases, the distribution of timescales broadens). Such distribution is similar to previously obtained results[13] and describes the broad distribution of inter-related timescales inherent to the viscoelastic heterogeneities of the different microtissue components.

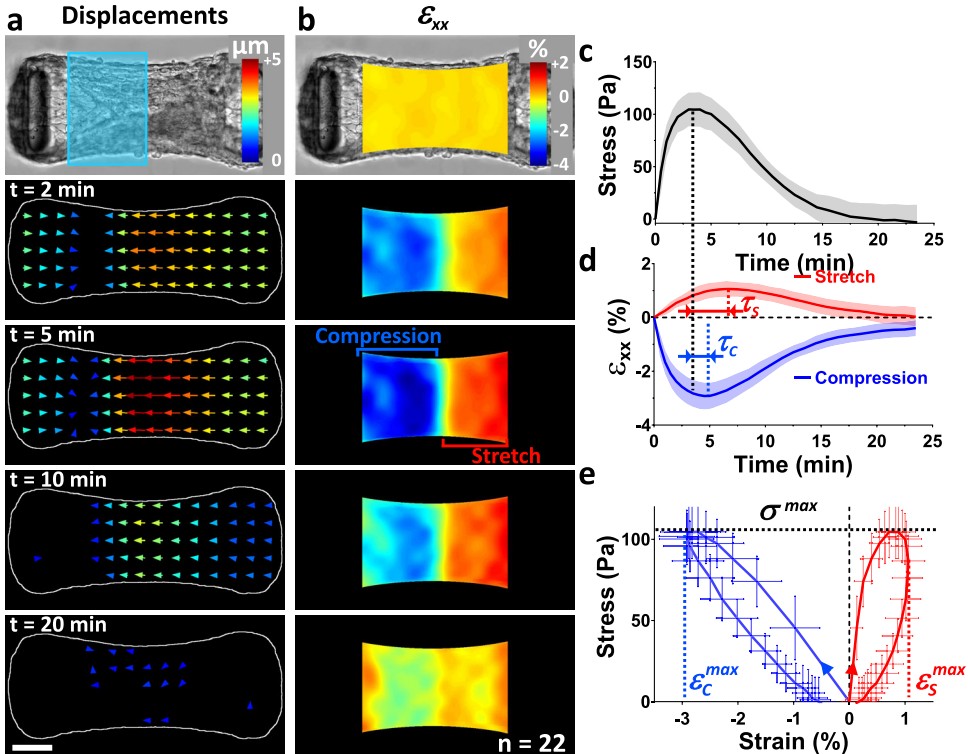

**Fig. 3 | Light-induced local contractions evidence the viscoelastic properties of microtissues.** Temporal evolution of the average displacement (**a**) and $\varepsilon_{xx}$ strain field (**b**) after stimulation of the left half of the tissue by blue light (symbolized by a blue rectangle in **a**). **c** Evolution of the stress $\sigma_{xx}$ over time and **d** resulting strain $\varepsilon_{xx}$ decomposed in its positive (stretch, in red) and negative (compression, in blue)

components. **e** Corresponding stress-stretch (in red) and stress-compression (in blue) curves. The dotted lines indicate the maximum stress $\sigma^{max}$ (in black), the maximum stretch $\varepsilon_S^{max}$ (in red) and the maximum compression $\varepsilon_C^{max}$ (in blue). Data are the average of 22 microtissues over 2 independent experiments ±SD. Scale is 100 μm. Source data are provided as a Source Data file.

We then tested the robustness of our approach by assessing the rheological properties of microtissues under different conditions (Supplementary Fig. 3). We first investigated the influence of the boundary conditions on the rheological properties of the microtissue. To this end, we stimulated locally microtissues suspended between cantilevers presenting a spring constant between 0.20 to 1.12 N/m, which have been previously shown to influence tissue contractility and stiffness[2,4,9]. We measured the resulting stretch of the non-stimulated zones and inferred elastic moduli and temporal delay between maximum stress and stretch. In good agreement with previous measurements using various spring constants[9], we found that the elastic modulus increased from 6.2 ± 2.5 to 21.4 ± 8.3 kPa with increasing cantilever spring constant (Supplementary Fig. 3).

We also observed that the delay $\tau_S$ between maximum stress and stretch more than doubled, from 133 ± 54 s to 282 ± 75 s, when the cantilever spring constant was increased from 0.20 to 1.12 N/m. As cellular contractility is the main regulator of tissue tension, while collagen architecture was shown to impact predominantly the tissue stiffness[9,11,12,27], our results suggest that this delay $\tau_S$ is due to the rupturing and reforming of bonds within the cytoskeleton (i.e. between actin filaments, between actin and myosin or between cadherin and catenin), in coherence with the previously shown stretch-induced perturbation of the cytoskeleton[13,28–30]. To test this hypothesis, we interrogated the dependence of the measured elastic modulus and time delay on the amplitude of the light-induced contraction. As the photo-activation of CRY2 depends on both the amplitude and the frequency of the light pulses[22,31], we varied the amplitude of contraction by varying either the irradiance of a single light pulse, or the number of 1 minute-spaced light pulses (Supplementary Fig. 4). We measured no significant differences in elastic modulus, while the delay $\tau_S$ between maximum stress and $\varepsilon_{xx}$ strain increased with the amplitude of contraction. We further verified that the light-induced

contraction was not inducing plastic, irreversible deformation of the collagen and/or actomyosin network. To this end, we illuminated the left half of microtissues with six successive light stimulations, spaced far enough apart in time to allow complete relaxation. We did not measure any difference in elastic modulus or time delay between the different stimuli (Supplementary Fig. 5). Altogether, these results demonstrate that using the cells themselves as mechanical actuators allows to probe, as the cells naturally do, reversibly and non-destructively, the elasticity of fibrous microtissues. Moreover, this approach evidences the stress-dependent dynamic of strain propagation in such tissues.

We then assessed the mechanical properties of microtissues 24 h and 48 h after seeding. We observed an almost doubling of the elastic modulus from 15.1 ± 6.5 kPa to 28.5 ± 11.1 kPa over these 24 h, while the change in the delay $\tau_S$ was not significant (Supplementary Fig. 3). Since collagen was shown to be a key component of the tissue stiffness[9], we also assessed the rheological properties of microtissue composed of different collagen densities. We measured a very slight increase of stiffness from 15.1 ± 6.5 to 19.0 ± 7.0 kPa for initial collagen densities varying between 1.5 and 2.5 mg/mL, once again without any significant change of viscosity (Supplementary Fig. 3). These values correlate with previous measurements[9] and the small variation of stiffness despite the large difference in initial collagen density is coherent with the insensitivity of stiffness to density for collagen under tension[15].

However, since collagen density feedbacks over time to regulate cell forces and tissue stiffness[9], we made good use of our non-destructive approach to assess the possible rapid changes of tissue stiffness and viscosity upon collagenase treatment. After 10 min of culture with 50 μg/mL of collagenase type I, we measured a strong decrease in tissue stiffness from 21.4 ± 8.3 to 9.1 ± 6.0 kPa, while the time delay between maximum stress and stretch was exactly the same before and after collagenase treatment. These results thus

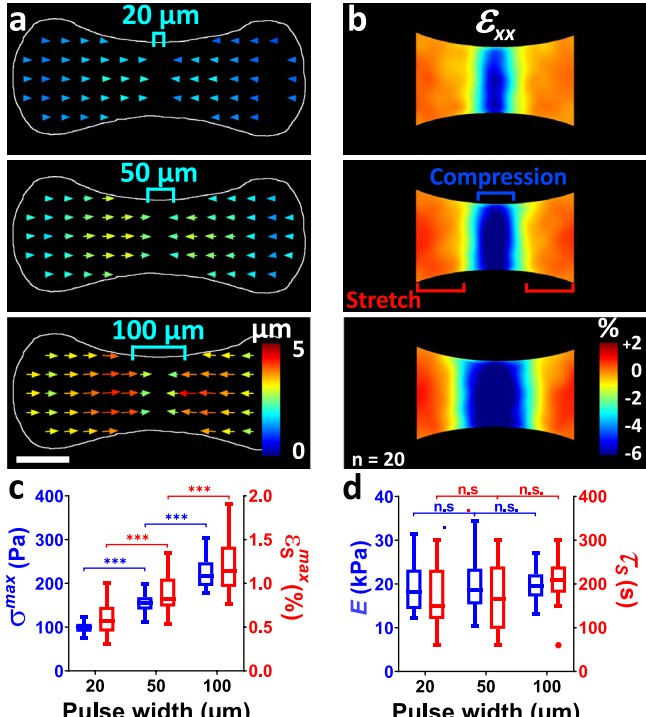

**Fig. 4 | Light-driven probing of the apparent elastic modulus of microtissues.**
Average displacement (**a**) and $\varepsilon_{xx}$ strain (**b**) fields resulting from 20 μm, 50 μm and 100 μm wide light stimulations (symbolized by blue brackets). For readability reasons, only half of the displacement vectors are represented. Scale bar is 100 μm. **c** Corresponding maximum stress $\sigma^{max}$ and stretch $\varepsilon_S^{max}$. **d** Resulting elastic modulus E, reported as $\sigma^{max}/\varepsilon_S^{max}$, and time delay $\tau_S$ between $\sigma^{max}$ and $\varepsilon_S^{max}$ in function of the pulse width. Data are presented as Tukey box plots (i.e. the box extends from the 25th to 75th percentiles, the median is plotted as a line inside the box and the whiskers extend to the most extreme data point that is no more than 1.5 times the interquartile range from the edge of the box) with n = 20 microtissues over 2 independent experiments. ***$P < 0.001$ between conditions. n.s. stands for non-significant (i.e. $P > 0.05$). Statistical significances were determined by one-way analysis of variance (ANOVA) corrected for multiple comparisons using Tukey test. Source data are provided as a Source Data file.

demonstrate that the cleavage of the peptide bonds of collagen by the collagenase strongly impacted the collagen structural stiffness without affecting the microtissue viscosity.

Finally, we explored the possibility to track changes in tissue mechanics during fibrogenesis. To this end, we induced the differentiation of fibroblasts to myofibroblasts by incubating microtissues with TGF-β₁, a well-known, strong fibrogenic growth factor[9,32,33]. We first confirmed the differentiation of opto-RhoA fibroblasts into opto-RhoA myofibroblasts by observing a strong increase of α-SMA stress fibers after 3 days of treatment with TGF-β₁ (Supplementary Fig. 6.a-c). We then assessed the elastic modulus E and the delay $\tau_S$ by optogenetics after 24 h and 72 h of culture with or without TGF-β₁. Similarly to previously reported measurements[6,9,33], we observed a strong increase of tissue stiffness for TGF-β₁-treated microtissues at day 3 (Supplementary Fig. 6d). We also observed that the delay $\tau_S$ between maximum stress and stretch is almost twice as long with versus without TGF-β₁ after 72 h (Supplementary Fig. 6e), which either supports a mechanism of stretch-induced perturbation of the cytoskeleton similar to the one we observed when varying the cantilever spring constant, or suggests a different relaxation dynamic of myofibroblasts versus fibroblasts.

Altogether, these results confirm the major role of collagen structure in the global stiffness of microtissues, whereas microtissue viscosity is mostly regulated by the contractile prestress of cells. As the

collagen structure depends on the microtissue geometry[26], we next sought to probe local mechanical anisotropies using our optogenetic approach.

## Optogenetic assessment of local anisotropies

As microtissues suspended between two cantilevers exhibit a pronounced, homogeneous anisotropy along the x-axis (Fig. 2), we generated microtissues suspended between four cantilevers, in order to induce a more heterogeneous architecture presenting various degrees of anisotropy[26]. We then performed a discoidal, isotropic light stimulation either in the center or on the left side of these square microtissues (Fig. 5a, b, Supplementary movie 6-7). We thus observed a completely isotropic displacement field for the centered stimulation, with a mean angle of 44 ± 4 °, whereas the eccentric stimulation induced a slightly biased displacement field with a mean angle of 53 ± 4° and more than 45% of the displacements oriented within less than 30° of the y-axis (Fig. 5c–e, Supplementary movie 6-7). These fields of displacement were paralleled with either an isotropic strain field for the centered stimulation, i.e. $\varepsilon_{xx} = \varepsilon_{yy} = -1.6 \pm 0.2\%$, or an anisotropic strain field for the eccentric stimulation with $\varepsilon_{xx} = -0.9 \pm 0.2\%$ while $\varepsilon_{yy} = -1.6 \pm 0.3\%$ (Fig. 2f–j, Supplementary movie 6-7). We thus obtained an A.C. of 0.0 ± 0.1 for the center part of the tissue whereas its left side exhibited an A.C. of −0.3 ± 0.1, indicating a moderate anisotropy along the y-axis (Fig. 5k), similarly to previous results obtained utilizing finite element models[26].

We then sought to compare these local anisotropic properties to the structural organization of these square microtissues. We simultaneously stained actin and collagen and quantified their respective orientations (Fig. 5l–q, Supplementary movie 8). In good agreement with previous studies[23,26], we found a mostly random organization of actin and collagen fibers in the center of the microtissues, while 65% of the actin fibers and 56 % of the collagen aligned within 30° of the y-axis along the left side. Bose et al. previously demonstrated that both ECM fiber alignment and density, resulting from the history of tissue formation, influence the local tissue stiffness[26]. The correlation between the strain pattern, probed via optogenetics, the fine-scale cytoskeletal and extracellular architecture, and possibly stiffness heterogeneities thus demonstrates that the spatial propagation of mechanical signals in microtissues is strongly dependent upon actin and collagen organization, which in turn depends on the formation history of the tissue.

Finally, we investigated whether optogenetics could also be used to influence this architecture and the resulting local contractility. We conditioned the top half of a square microtissue with repetitive stimulations every 5 min for the first 24 h of its formation, before probing its contractility with a single, narrow, perpendicular pulse (Supplementary Fig. 7a, b). We thus observed a larger compression of the conditioned top half, reaching 3.0 ± 0.5%, compared to a 2.1 ± 0.7% compression of the non-conditioned bottom half (Supplementary Fig. 7c–e). The top half of the microtissues also exhibited a slightly higher signal of actin fluorescence (Supplementary Fig. 7f–h) but no difference in collagen fluorescence (Supplementary Fig. 7i–k) after immuno-staining. Altogether, these results suggest that, similarly to long term, external mechanical stimulations, a long term optogenetic conditioning during tissue formation impacts the expression and/or maturation of the actomyosin machinery[11,34], with the additional ability to do it locally.

Consequently, the use of optogenetically-modified cells as actuators to probe their environment appears as a powerful tool to guide tissue formation while simultaneously probing non-destructively architectural heterogeneities.

## Discussion

The mechanical properties of microtissues have been identified as major factors that regulate the physical integrity, contractility and function of such microtissues[3–5,9]. Yet, these properties have only been

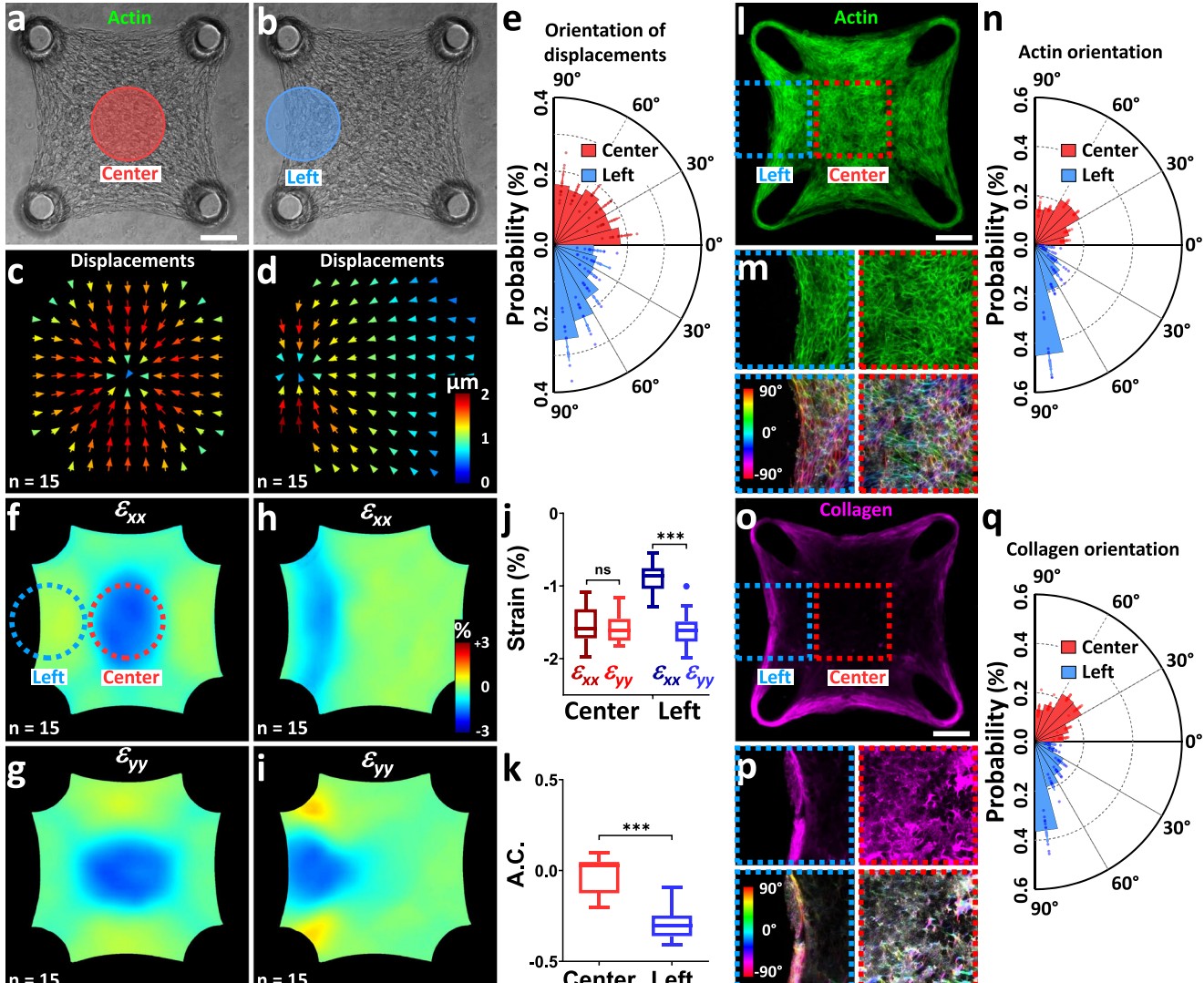

**Fig. 5 | Optogenetically-mapped mechanical heterogeneities correlate with actin and collagen architecture. a** Representative square microtissue where the red (**a**) and blue (**b**) discs represent the centered and eccentric 200 μm diameter stimulation areas, respectively. Corresponding average displacement fields for the centered (**c**) and eccentric (**d**) stimulations. For readability reasons, only half of the vectors are represented. **e** Polar plot of the orientations of the displacement vectors. Data are presented as mean ± SD with $n = 15$ microtissues over 2 independent experiments, superimposed with a dot plot of the data distribution. Resulting average strain fields for centered ($\varepsilon_{xx}$ in **f**, $\varepsilon_{yy}$ in **g**) and eccentric ($\varepsilon_{xx}$ in **h**, $\varepsilon_{yy}$ in **i**) stimulations. Comparison of the average strain amplitudes (**j**) and anisotropy coefficient A.C. **k** in the areas of stimulation. Data are presented as Tukey box plots (i.e. the box extends from the 25th to 75th percentiles, the median is plotted as a line inside the box and the whiskers extend to the most extreme data point that is no more than 1.5 times the interquartile range from the edge of the box) with $n = 15$ microtissues over 2 independent experiments. ***$P < 0.001$ and n.s. stands for nonsignificant (i.e. $P > 0.05$), determined by two-tailed $t$-test. Confocal projection of the fluorescent staining of actin in a representative microtissue (**l**), magnifications and color-coded map of the actin orientation (**m**) and polar plot (**n**) of the orientations of actin fibers in the center (red) or the left side (blue). Confocal projection of the fluorescent staining of collagen in a representative microtissue (**o**), magnifications and color-coded map of the collagen orientation (**p**) and polar plot (**q**) of the orientations of collagen fibers in the center (red) or the left side (blue). Data in (**n**) and (**q**) are presented as mean ± SD with $n = 16$ microtissues over 2 independent experiments superimposed with a dot plot of the data distribution, and the probability of alignment 75°–90° between the center and left conditions are significantly different: ****$P < 0.0001$ determined by two-way ANOVA corrected for multiple comparisons using Sidak test. Scale bars are 100 μm. Source data are provided as a Source Data file.

examined through the application of external loading and biochemical treatments[6,11,33], which yielded key insights into how cells and ECM architecture impact microtissues' mechanics. However, external actuations are inherently different from cell-generated forces, and the low spatial and temporal resolution of biochemicals limits their use for studying how cells probe the mechanics of their environment and how these mechanical perturbations propagate within the surrounding tissue. By introducing optogenetic modulation of the cellular contractility to the microtissues, we were able to produce local, physiological, cell-generated mechanical perturbations while simultaneously tracking their spatial and temporal propagation. These perturbations

were reversible and paralleled by a compression of the stimulated zone, while non-stimulated zones were stretched.

We thus used optogenetic cells as biological actuators to probe the mechanics of fibroblasts/collagen microtissues. By tracking the maximum amplitudes of stress, compression and stretch in response to local light-activation of the contractility, we were able to plot hysteretic stress/strain curves, as the strains were delayed in time compared to the stress. We first observed that these fibrous microtissues seemed more compliant in compression than in stretch, as the compressed, stimulated part was more deformed than the stretched, non-stimulated one. Previous work has shown the highly nonlinear

behavior of collagen gels, with a lower stiffness in compression compared to stretch[35,36], and appropriate modeling of our light-sensitive microtissues would be particularly relevant to characterize this mechanical behavior in more detail. We further demonstrated that, within the range of stimulation area and intensity tested in this work, single light-induced contractions did not induce plastic deformation of the microtissue. However, the amplitude of the contraction, modulated via the light-stimulation or the differentiation of fibroblasts into myofibroblasts, impacted the delay $\tau_S$ between stress and strain. These results thus evidence a stress-dependent dynamic of strain propagation in fibrous microtissues, similarly to previous work in fibroblast populated collagen matrices that described an increase of the loss modulus with the strain rate[27] or an increase in stress-strain hysteresis with the strain rate[37], or in presence of myofibroblasts[33]. One hypothesis for this behavior is a dependence between amplitude and dynamics of stress. However, we did not measure any difference in the time from zero to maximum contraction stress when varying the light intensity or the cantilever spring constant. A second hypothesis is an inherent stress-dependent viscosity of the collagen matrix, but the inexistent difference in $\tau_S$ when the collagen density is varied or when the collagen is cleaved with collagenase proves otherwise. A third hypothesis is the catch bond behavior of the contractile cytoskeleton that may induce such stress-dependent strain propagation. Although the catch-bond behavior of actin-myosin or cadherin-catenin bonds was demonstrated in single molecules[28,30] or single cells experiments[29], our results motivate further experimental studies to test whether this hypothesis is indeed correct in fibrous tissues, possibly via the use of cytoskeleton-associated tension reporters[38–40].

Using this optogenetic approach, we found that boundary conditions affect the stiffness of microtissues and, because of their impact on cell-generated stress[2,9], the delay between stress and strain, whereas collagen density or its degradation by collagenase only impact tissue stiffness. In the long term, we showed that tissue stiffness increases over maturation time, especially in the case of fibrogenic maturation, which correlates well with a cell-driven cross-linking of the matrix during tissue maturation or fibrosis[33,41,42].

These results are in good agreement with previous work showing that, in the short term, the collagen matrix is the main determinant of the tissue stiffness[9,11] while the actomyosin cytoskeleton is predominantly responsible for the viscoelasticity of microtissues[13,43].

Furthermore, we showed that the orientations and magnitudes of the light-induced contractions correlated with the collagen and actin orientation, which is known to influence in turn local tissue stiffness[26], and that repetitive stimulations during tissue formation could influence the expression and/or maturation of the actomyosin cytoskeleton, highlighting the predominant role of fibrillar architecture in the spatial propagation of mechanical signals. As collagen organization has been shown to be key to the long-scale interactions between cells and their navigation through the ECM[44–46], a more precise characterization using second-harmonic microscopy would be key to delineate the respective roles of thin isolated fibers and thick, strongly anisotropic collagen bundles.

Moreover, although we observed a correlation between cell-induced strain patterns and tissue architecture, we cannot deduce from our experiment the cause of the anisotropic strain pattern, whether it is due to the cytoskeletal and extracellular organization, the spatial heterogeneity in stiffness or a combination of both parameters. Microtissue regions with aligned actin and collagen fibers were shown to be significantly stiffer than randomly organized regions[26], but unraveling the respective roles of the different types of anisotropy (i.e. anisotropy of cellular contraction, mechanical properties, fiber orientations) remains currently an extremely complex task that has only begun to be undertaken, and would require further experimental studies, possibly combining tissue engineering, optogenetics, cytoskeleton-associated tension reporters and computational modeling[23,26,38–40]. Nevertheless, the ability of our method to apply dynamic, internal mechanical perturbations, while simultaneously measuring the magnitude and spatio-temporal propagation of such perturbations opens an exciting avenue for guiding the formation of tissues while simultaneously assessing their architectural heterogeneities, as well as for examining the mechanical guidance of cell migration in fibrillar matrices.

However, as our approach uses the contraction of one part of the tissue (the stimulated one) to stretch the other part (the non-stimulated one) to probe the mechanical properties of the latter, it inherently requires an active actomyosin machinery, which could complicate the study of the role of cytoskeletal components in regulating tissue mechanics. Indeed, complete inhibition of the contractile ability of the microtissue, using high doses of actomyosin-targeting drugs for example, would hinder the use of our approach. Yet, our method remains valid as long as the contractile machinery is at least partially active, e.g. for small to medium doses of actomyosin-targeting drugs. Similarly, although the recruitment of CRY2 to the membrane is very fast (few seconds), its dissociation is slower (several minutes)[22], while the dynamic of the RhoA pathway leads to a contractility activation in the order of tens of seconds and a slower relaxation of several minutes[19]. As a result, the induced cell contraction and relaxation takes almost 20 min, thus precluding its use for probing rapid mechanical changes or quantifying time-dependent mechanical responses of microtissues at different frequencies. Finally, our optogenetic target RhoA has many downstream effectors. Although we used single, short stimulations that induced an elastic response of the microtissue with no long lasting effects such as cytoskeleton remodeling for example, we cannot exclude that off target effects could exist, via for example the self-amplification and self-inhibition of RhoA[47].

In conclusion, the combination of tissue engineering and optogenetics provides unique opportunities to quantitatively demonstrate the impact of physical and biological parameters on the generation, propagation and sensing of cell-generated mechanical perturbations in 3D tissues. Most importantly, our approach paves the way to probing the rheology of 3D tissues in real time and non-destructively, using their own constituting cells as internal actuators. These same attributes will likely provide valuable opportunities to elucidate how mechanical cues dynamically regulate tissue formation and function over space and time. Altogether, our work thus demonstrates that an optogenetic control of cell contractility, combined with specifically designed microtissue geometry and possibly computational modeling[26], could offer a powerful approach to analyze the complex interplay between mechanical boundary constraints, cell contractility, ECM density, alignment, and mechanical properties.

## Methods

### Cell culture, reagents and immuno-stainings

The DHPH domain of ARHGEF11 Guanine Exchange Factor gene was cloned into CRY2PHRmCherry using Nhe1 and Xho1 cloning sites. ARHGEF11-CRY2PHR-mCherry (AddGene #89481) and CIBN-GFP-CAAX (AddGene #79574) were then inserted into lentiviral back-bones (pHR and pLVX respectively) to create the stable cell line of opto-RhoA fibroblasts from NIH 3T3 fibroblasts (#CRL-1658, ATCC). After viral transduction, only the top 2% of the cells presenting the highest expression levels for both transgenes were FACS-sorted before amplification. Gates were chosen such that fluorescence levels in GFP and mCherry channels were approximately 100 times higher than in control WT cells. The obtained opto-RhoA fibroblasts[20,48] (<15 passages, kindly provided by L. Valon and M. Coppey, Institute Curie, Paris, France) were cultured in Dulbecco's modified Eagle's medium (DMEM, Gibco Invitrogen) supplemented with 10% fetal bovine serum (FBS, Gibco Invitrogen), 100 U/ml of penicillin and 100 μg/ml of streptomycin (Gibco Invitrogen), and kept at 37 °C in an atmosphere

saturated in humidity and containing 5% CO2. Collagenase experiments were performed by incubating microtissues with 0.05 g/L of collagenase type I (Sigma) for 10 min before thorough rinsing with PBS and replacement with growth medium. To induce myofibroblasts differentiation in the microtissues, regular culture media was supplemented with 5 ng/mL of TGF-$\beta_1$, (T7039, Sigma) just after cell seeding. Prior to immunostaining, samples were fixed in 4% paraformaldehyde (Sigma) and blocked with 2% BSA (Sigma). Collagen and α-smooth muscle actin (α-SMA) were immuno-stained with a monoclonal anti-collagen type I antibody produced in mouse (clone COL-1, Sigma-Aldrich #C2456) and a monoclonal anti-α-smooth muscle actin produced in mouse (clone 1A4, Sigma-Aldrich #A2547), respectively, both diluted 1:200 in tris-buffered saline (TBS, Sigma). Both antibodies were detected with goat anti-mouse IgG (H + L) cross-adsorbed secondary antibody, Alexa Fluo 647 (Invitrogen #A-21235) diluted 1:200 in TBS. Samples were permeabilized with 0.5 % Triton X-100 (Sigma) in TBS either before or after the incubation with the primary antibody, in order to stain for intracellular α-SMA or extracellular collagen I, respectively. Actin was labeled with phalloidin-Atto488 (Sigma) and nuclei with DAPI (Invitrogen).

### Device fabrication and calibration
The microtissues are engineered in microwells containing two or four T-shape cantilevers. This T-shape is essential to constrain the microtissue formation, ensure good anchorage of the microtissue to its supporting cantilevers and avoid its slipping from the post. To achieve this complex geometry with a top cap wider than its post, SU-8-based masters were fabricated following the technique described previously[2,49]. Briefly, successive layers of negative and positive photoresist (Microchem) were spin coated, insolated and baked to create multilayers templates. A first layer of negative resist allows the creation of the posts, a second one consisting of a mix of 70 % negative and 30 % positive photoresist serves as a lithographic-stop layer that prevents unwanted cross-linking of the underlying layer, and a third layer of negative photoresist leads to the creation of the top wide cap at the tip of the post.

Polydimethylsiloxane (PDMS, Sylgard 184, Dow-Corning) microfabricated tissue gauges (μTUGs) were molded from the SU-8-based masters by double replication as described previously[2,49]. Briefly, SU-8-based masters or PDMS replicates were oxidized in an air plasma, silanized with trichloro(1H,1H,2H,2H-perfluorooctyl)silane (Sigma) vapor overnight under vacuum to facilitate subsequent release of PDMS from the template and avoid tearing the top cap of the cantilevers. Prepolymer of PDMS was then poured over the template, degassed under vacuum, cured at 65 °C for 20 h, and peeled off the template. Different ratios of PDMS/curing agents were used to modulate PDMS stiffness, which was assessed through uniaxial extension of 50 x 4 x 1 mm strips with an Instron 5848 Microtester (Instron). Stiffness was determined from the linear region of the obtained stress-strain curves. Cantilever spring constants were calibrated with a capacitive MEMS force sensor mounted on a micromanipulator as described previously[4,23]. Briefly, the sensor tip was placed 20 μm below the top of the cap and the probe translated laterally against the outer edge of the cantilever. The spring constant was calculated from the displacement of the cantilever head and the reported sensor force. The spring constant of the cantilevers was found to be $0.20 \pm 0.03$ N/m, $0.45 \pm 0.10$ N/m and $1.12 \pm 0.26$ N/m for PDMS/curing agent of 1:20, 1:10 and 1:4, respectively. The dimensions and geometry of the cantilevers were regularly assessed to ensure they are not affected by successive replications.

Before cell seeding, the PDMS templates were sterilized in 70% ethanol followed by UV irradiation for 15 min and treated with 0.2% Pluronic F127 (Sigma) for 2 min to reduce cell adhesion. A reconstitution mixture, consisting of 1.5 mg/mL or 2.5 mg/mL liquid neutralized collagen I from rat tail (Advanced Biomatrix) was then added to the surface of the substrates on ice and templates were degassed under vacuum to remove bubbles in the liquid. A cooled suspension of 750,000 cells within reconstitution mixture was then added to the substrate and the entire assembly was centrifuged to drive the cells into the micropatterned wells, resulting in approximately 500 cells per well. Excess collagen and cells were removed by de-wetting the surface of the substrate before incubating at 37 °C to induce collagen polymerization for 9 min. Culture medium was then added to each substrate. Microtissues were kept in the incubator for 24 h before stimulation experiments, unless specified otherwise. Over these first 24 h of cultivation, the fibroblasts spread and spontaneously compact the collagen matrix (Fig. 1). The two T-shaped cantilevers anchor and constrain the contraction of the collagen matrix to form a microtissue that spans across the top of the pair of cantilevers.

### Optogenetic stimulation and microscopy
Optogenetics stimulation and brightfield imaging were performed using an inverted Nikon Eclipse TI-2 microscope with an Orca flash 4.0 LT digital CMOS camera (Hamamatsu) and a CFI S Plan Fluor ELWD 20x/0.45 objective (Nikon). Light stimulations were achieved using a Mosaïc 3 digital micromirror device (DMD, Andor) and the 470 nm diode of a Spectra X light source (Lumencor). The light power was calibrated using a photodiode power sensor S120C (Thorlabs), placed in the light path, after the objective, and connected to a compact power and energy meter console PM100D (Thorlabs). The total irradiance sent to the microtissue could be varied from 0 to 4.8 mW/mm². Microtissues were stimulated with an irradiance of 1.8 mW/mm² for 500 ms for all the experiments, unless specified otherwise. The light pattern, intensity and duration, as well as the microscope itself, were controlled using NIS Elements software (Nikon). Microtissues were maintained at 37 °C and 5% CO2 in a top-stage incubator (Oko Lab).

Confocal images were obtained with a Leica laser scanning microscope (LSM SP8, Leica) equipped with a plan apochromatic 40x/1.30 objective (Leica). Actin and collagen orientations were evaluated from confocal z-stack images with the Orientation-J plug-in (http://bigwww.epfl.ch/demo/orientationj/)[50] in Image J. This plugin computes the structure tensor for each pixel in the image by sliding a Gaussian analysis window (variance σ = 2 pixels) over the entire image. From the structure tensor are extracted both the orientation and coherency properties of the region of interest. These properties are then gathered in a color map in HSB (Hue Saturation Brightness) mode where the hue corresponds to the orientation, the saturation to the coherency and the brightness to the source image. The coherency indicates if the local image features are coherently oriented or not. The polar histograms presented in Fig. 2 and Fig. 5 are weighted histograms, the weight being the coherency. Consequently, thicker or denser collagen bundles are weighted more than isolated, randomly oriented fibers, in coherence with the fact that thick and dense collagen bundles weight more in the mechanical behavior of a fibrous microtissue than thin random fibers[26].

### Force measurement and strain measurement
The contraction force generated by individual microtissues before, during and after light stimulations was assessed from the deflection of the cantilevers. This deflection $d$ was determined by comparing the position of the top of the cantilevers to their initial position (i.e. before seeding the cell/collagen mixture). To this end, brightfield images were taken every 30 s and the displacement of the top of the cantilevers was tracked using a custom MATLAB script. Briefly, the displacement of the sharp contrast created by the edge of the cantilever heads was tracked over time by auto-correlation of the interpolated line profiles, allowing subpixel resolution. Tracking results were visually checked and faulty tracking discarded. Based on the linear bending theory, the resulting force $F$ applied on a cantilever was inferred from its deflection and its

spring constant $k$ as follows: $F = k.d$. The force generated by one microtissue corresponds to the average force applied to each of its two anchoring cantilevers. Only tissues that were uniformly anchored to the tips of both cantilevers were included in the analysis. This anchorage between microtissue and cantilevers was visually checked over the whole duration of the experiment. Only microtissues wrapping completely the cap of both cantilevers were selected (Supplementary Fig. 8a–e). Tissues tearing or slipping from their cantilevers during an experiment were discarded from the analysis.

The stress $\sigma_{xx}$ was calculated by dividing the force by the cross-sectional area measured in the center of the tissue. The width of each microtissue (i.e. its dimension along the y-axis, the x-axis being between the two cantilevers) was tracked during stimulation experiments. The cross-sectional areas were measured from confocal z-stack images of tissues fixed and stained right after experiments, in their center, similarly to previous works (Supplementary Fig. 8f–h)[2,4,6,9,11,26]. No measurable change in cross-sectional area was observed over the duration of light-stimulation experiments, in agreement with the almost inexistent displacements along the y-axis (Figs. 1–4). Consequently, the cross-sectional areas were considered unchanged over the duration of stimulation. Of note, the cross-sectional areas were linearly correlated with the tissue width squared $w^2$ (Supplementary Fig. 8h, $R^2 = 0.80$), which allowed estimation of the tissue cross-section when microtissues could not be fixed immediately after experiments.

The displacement fields were determined from the brightfield images using a particle image velocimetry (PIV) algorithm implemented as a Matlab toolbox (https://pivlab.blogspot.com/)[51]. Briefly, small sub images (interrogation areas) of an image pair consisting of the reference image at $t = 0$ and the image of interest were cross-correlated in the frequency domain using FFT to derive the most probable particle displacement vector in the interrogation areas, with its $x$ and $y$ components $u$ and $v$, respectively. Analyzed images were $1024 \times 512$ pixels ($800 \times 400\,\mu m$) or $1024 \times 1024$ pixels ($800 \times 800\,\mu m$) for microtissues between 2 or 4 cantilevers, respectively, and the size of the interrogation areas was successively reduced from $128 \times 128$ to $96 \times 96$ and finally $64 \times 64$ pixels (corresponding to $100 \times 100$, $75 \times 75$ and $25 \times 25\,\mu m$, respectively). For each pass, i.e. for each size of interrogation area, the overlap between interrogation areas was set to 50%, leading to a final resolution of $25\,\mu m$. Outliers were filtered using a local normalized median filter[52], missing vectors were replaced by interpolated data[53] and the noise was reduced using a penalized least squares method[54].

The strain maps were derived by numerical differentiation of $u$ and $v$ to both $x$ and $y$, comparing the displacement vector of each particle with the vectors of surrounding particles in a window of $3 \times 3$ particles[55]:

$$\varepsilon_{xx} = \frac{\partial u}{\partial x}; \varepsilon_{yy} = \frac{\partial v}{\partial y}; \varepsilon_{xy} = \frac{1}{2}\left(\frac{\partial u}{\partial y} + \frac{\partial v}{\partial x}\right) \qquad (1)$$

For plotting stretch and compression over time, the $\varepsilon_{xx}$ strain was estimated by averaging the displacement $u$ (i.e. along the x axis) over the width of the tissue before calculating its slope across the length of the stimulated and the non-stimulated areas, respectively. The tissue was considered as homogeneous and was not discretized as individual cells and matrix. Consequently, we measured tissue displacements and corresponding tissue deformations. Of note, cell migration over the duration of a contraction was considered negligible compared to cell deformation.

## Statistics and reproducibility

For each line graph, the mean represents the mean±standard deviation of $n$ microtissues ($n$ is defined in the caption of each figure). For each box plots, the box extends from the 25th to 75th percentiles, the median is plotted as a line inside the box and the whiskers extend to the most extreme data point that is no more than 1.5 times the inter-quartile range (IQR) from the edge of the box (Tukey style). Significances were assessed with Prism (GraphPad). Exact $P$-values are provided in the source data file.

## Reporting summary

Further information on research design is available in the Nature Portfolio Reporting Summary linked to this article.

## Data availability

Source data are provided as a Source Data file. Because of the large file size, raw image data are available from the corresponding author on request. A response will be provided in less than 2 weeks. Source data are provided with this paper.

## Code availability

The Matlab analysis procedure to track for pillar displacements is available on GitHub (https://github.com/Orion38/Pillar-tracker).

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

## Acknowledgements

A.M. acknowledges support from the University Grenoble Alpes Ph.D. fellowship program. M.B. acknowledges financial support from the ANR MechanoSwitch project, grant ANR-17-CE30–0032-01, the ANR Platfor-Mech project, grant ANR-18-CE14–0037-02, the ANR Inter-s-cal project, grant ANR-21-CE13–0042-02. G.C. acknowledges financial support from the ANR SupraWaves project, grant ANR-19-CE13-0028 of the French Agence Nationale de la Recherche (ANR). T.B. acknowledges funding

through CNRS grants (Actions Interdisciplinaires 2017, DEFI Instrumentation aux limites 2017, Tremplin@INP 2021, PEPS CNRS-INSIS 2021, Lumière Visible et Vie 2022). This work was supported by the Center of Excellence of Multifunctional Architectured Materials "CEMAM" (n° AN-10-LABX-44-01). The authors thank P. Moreau and I. Wang for their technical support, L. Vallon, S. de Beco and M. Coppey for kindly providing the opto-RhoA fibroblasts, G. Chagnon and N. Briot for the mechanical characterization of the PDMS, as well as P. Recho for helpful discussions.

## Author contributions

A.M., A.R., G.C., M.B. and T.B. conceived the study and designed the experiments. A.M., A.R., J.R. and T.B. performed experiments and data analysis. A.M. and T.B. wrote the manuscript with feedback from all authors. M.B., G.C. and T.B. supervised the project.

## Competing interests

T.B. is co-inventor with Christopher S. Chen, Ken Margulies, Wesley Legant and Michael T. Yang of the active patent #US9512396B2 entitled "In vitro microphysiological system for high throughput 3D tissue organization and biological function", relating to the fabrication and use of the microsystem utilized in this manuscript. A.M., A.R., J.R., M.B. and G.C. declare no competing interests.
