## [Peer Review File · Nature Communications]

REVIEWER COMMENTS

Reviewer #1 (Remarks to the Author):

The work of Méry et al. reports on a method for measuring mechanical properties of tissues based on the light-controlled stimulation of tissue contraction, with simultaneous recording of tissue stress and strain. The method uses opto-genetically modified fibroblasts cultured with collagen that form microtissues suspended between opposing microcantilevers, allowing for direct measurements of tissue tension. Optical stimulation of these fibroblasts causes RhoA-activation and subsequent cell contraction. The assay is further compatible with fluorescence microscopy to monitor tissue flows and architecture. Using this setup, the authors describe how microtissues locally contract and stretch using measurements of tissue tension and particle image velocimetry.

Using cells as mechanical actuators to probe tissue material properties is an interesting concept (as opposed to perturbing tissues externally) and the authors provide clear proof-of-concept data for this approach. The novelty of the study is that it combines opto-genetic cell control and mechanical measurements on free-hanging microtissues, two methods that have been described independently before. The authors probe various geometries of microtissues and perform perturbations of collagen density and acute collagen digestion. Their obtained data confirm previously described viscoelastic tissue features (including the role of collagen in regulating tissue stiffness) and the stress-dependent alignment of cell/matrix structures.

As a weakness, the study does not present major new technological advancements or biological insight into the regulation of tissue mechanics. The overall capacity of the system seems to be not fully exploited and a detailed physical description or modelling approach has not been performed, which limits new physical and biological insight.

In addition, specific methodological aspects and data interpretations remain unclear and should be addressed:

- The quantitative analysis appears limited and does, for example, not address specific power law or exponential response behaviours associated with previously described viscoelastic tissue models and studies employing external perturbation.
- Strain was generated and assessed within a small range and the authors should clarify if this range can be extended and how a broader strain variation affects measured tissue parameters. Along this line, the authors vary illumination area but not the illumination time. Would an increase in the illumination time lead to a larger contractile stress or did local tissue contraction already saturate for the given settings? How would the variation of illumination time affect measured tissue parameters?
- Can local tissue areas be repetitively stimulated for contraction and how would this affect tissue mechanical properties? It would also be relevant to address if the use of cells as mechanical actuators can affect measured tissue properties differently compared to externally applied force measurements.
- The authors refer to previously published work how differences in cantilever stiffness can affect tissue stiffness and viscosity but this remains speculative if not explicitly tested experimentally.
- To extend the biological insight of the current study a specific perturbation of cytoskeletal elements, cell-matrix adhesion or mechanosensitive pathways could be performed to gain

insight into the regulation of tissue mechanics beyond previous reports.

Minor issues

- The basics of the culture technique should be more clearly described both in the schematic figure 1A and the methods section, in addition to referring to previously published work. It should be emphasized that the microtissue is free-hanging and clearly described how this is achieved, given that this has major relevance for the tension measurements described.
- Regarding the cell displacement analysis, the authors should make a clearer distinction between cell elongation versus cell displacement/flow and how it would influence tissue visco-elasticity.
- Analysis of actin/collagen alignment: How do the authors distinguish actin alignment from cell shape? Cortical actin appears very bright and cells are elongated along the x-direction. This could yield a cell alignment parameter rather than an actin fibre alignment. The resolution in Fig 2J further appears limited to analyse collagen alignment. The chosen colour bar for actin/collagen alignment also makes it difficult to distinguish the alignment in parallel and vertical direction (pink=0° and red 90° are similar). Also, what does the white colour indicate?

Figure 2: The authors argue that the anisotropic orientation of actin and collagen fibres are responsible for the anisotropic propagation of mechanical stimulation from an isotropic stimulus. How do the authors argue about the fact that the applied stimulus itself is not isotropic? (as it's not equidistant from the edges the force experienced by the different regions of the tissue will not be symmetric as in Figure 5)

Figure 3:

- It is not fully clear how stress (σ_{xx}) was calculated and a more detailed description in the method section would be beneficial ("deduced from the pillar deflection and the cross-section of the microtissue").
- Authors use ϵ_{xx} and "strain" interchangeably within the same figure, it would be easier to understand if this labeling was consistent

Figure 5: It is not clear whether collagen is present in the center of Figure 5O and the related supplementary movie 8, can the authors comment?

- Can the authors also clarify the reasoning of this sentence "In order to infer the rheology of the microtissue, we assimilated the stimulated part of the tissue to an actuator stretching the non-stimulated part."
- The authors could also more extensively discuss limitations of their presented approach, such as the need of keeping the actomyosin cytoskeleton intact to use cells as internal actuators, which complicates the study of the role of actin/myosin or actin binding proteins in regulating tissue mechanics (and a significant role has been attributed to the actomyosin cytoskeleton in previous work).

Reviewer #2 (Remarks to the Author):

Title: Light-driven biological actuators to probe the rheology of 3D microtissues

Authors: Adrien Méry, Artur Ruppel, Jean Révilloud, Martial Balland, Giovanni Cappello and Thomas Boudou

Journal / ID:

Summary: The authors present an interesting application of optogenetics to assess the mechanical behavior of microtissues through light-sensitive cell-driven contractions. The ability to characterize a tissue using intrinsic actuators could be quite valuable for both characterization of microtissues and better understanding the mechanisms of cellular behavior.

The study demonstrates the value of this method for assessing microtissue mechanical behaviors and studying how various factors can influence the structure and properties of the microtissues. The study is, however, limited in its verification of this method for measuring mechanical properties and incomplete in its assessment of the anisotropy of the microtissues.

General Comments

1. While the obtained elastic modulus is said to be in "very good agreement" with the results of prior studies, I note that different parameters were used in those studies. This study would be strengthened by performing a more direct verification, i.e. assessing the mechanical properties of a set of samples using both light-stimulated contractions and some other means (such as the magnetically actuated microcantilevers). This verification is very important for this manuscript, since the manuscript is focused on presentation of a method.

Additionally, some verification of the viscosity measurements, either through references or preferably direct comparisons should be provided.

2. Assessment of anisotropic behavior should more clearly delineate the different forms of anisotropy at play in this context (anisotropy of mechanical properties, anisotropy of cellular contractions, and anisotropy of fiber orientations). Figure 2 shows that strains in the x-direction are significantly greater in magnitude than strains in the y-direction, this behavior could be attributed to anisotropy of the elastic modulus and/or anisotropy of the cellular contractions, both of which could be related to the anisotropic orientation of actin and collagen. The two-cantilever set up, which is only capable of determining x-direction forces, thus seems ill-suited to fully analyze the anisotropy of the tissue.

3. The strain fields shown (e.g. Figure 2D and 2E) are astonishingly smooth. In this reviewer's experience, which is backed up by work by others (e.g. Midgett et al. *Acta Biomater.* 2017 Apr 15;53:123-139. doi: 10.1016/j.actbio.2016.12.054. Epub 2017 Jan 17), simply differentiating the displacement field numerically, as indicated in the section "Force measurement and strain measurement" results in noisy strain fields. Please provide more information on the method used, beyond simply citing reference 36, as the type of noise present in the images and displacement field depend strongly on the mode of imaging, contrast, and resolution.

4. Figure 3E appears to show higher stiffness in tension vs. compression. This is an interesting result in regards to the purported interest in comparing cell-generated forces to tissue stiffness. Some discussion of this finding would strengthen the manuscript.

5. In the 'Inferring mechanical properties of stretched areas' section, the authors report that "force increase was proportional to the area of the stimulated zones" and from that infer that light-induced contraction is proportional to the number of stimulated cells, but there is no accounting for the potential influence of cell density. Would a higher cell density and a smaller area of stimulation cause the same force increase as a proportionally lower density and larger area?

6. Given that part of the focus of the measurements is on time-dependence of the contraction and relaxation, I suggest the study also include multiple cycles of contraction. This would provide insight into whether the contractions are inducing more permanent matrix changes, or at least matrix changes that accumulate with repeated loading, whether this changes the viscosity, and how this affects the ability of the cells to continue inducing

tissue contraction.

Specific Comments

- 1. Introduction, 2nd paragraph, "it remains problematic to assess how such external mechanical stimulations compare with internal, cell-generated contractions":** This argument would be stronger if the manuscript provided a rationale for making this comparison. What is the biological implication?
- 2. Pg. 6, "This anisotropic contraction was paralleled with a strongly anisotropic field of deformation":** Force was only measured in one direction, so there is no way to know from the data collected if the forces are anisotropic; instead, we know only that the contractile deformation is anisotropic, which is the same thing as the deformation field being anisotropic. Thus, the two quantities described as "parallel" to each other are one and the same.
- 3. Pg. 11 – A doubling of the elastic modulus from 15.1 kPa to 28.5 kPa from 24-48 hours is reported, however on the previous page, an average elastic modulus of 19.2 kPa is reported. Since the 19.2 kPa measurement was presumably taken at 24 hrs, it's not clear why there's a difference between the 15.1 kPa and 19.2 kPa measurements.**
- 4. The figures clearly show that the cross-sectional area of the tissue is not constant, so some explanation of how this was accounted for in the calculation of stress is needed.**
- 5. Discussion, "whereas collagen density or integrity only impact tissue stiffness":** no measurement of collagen integrity is provided

Reviewer #3 (Remarks to the Author):

I believe the authors are correct in saying that optogenetics has not been used to control specific cells within a population and thereby control force in the tissue. The work is therefore significant and presents a potentially useful new tool. However, I believe there are some issues that need to be addressed:

The method that was actually used to induce cell contraction is published, so the novelty of the paper does not lie in developing this method, but in applying it in a new way. I believe there may be issues with the optoRhoA method as used here. Opto-RhoA functions by recruiting a Rho GEF to the membrane. Does this necessarily lead only to contraction? It is possible that the RhoGEF is completely specific for RhoA, but it would be valuable to have a discussion of the level to which this has been proven. More importantly, Rho activates and inhibits multiple downstream pathways, not just contraction. Formin activation in fact leads to protrusion. The experiments in the paper show that, regardless of what other effects are induced in this specific cell type and tissue, contraction is produced. Is this true in other cells and circumstances? Is there complex biology with some cells protruding and others contracting (e.g. cell versus edge of irradiated region), with net contraction. The authors have shown that in the right situation the technique can be useful, but I believe they should discuss caveats of the method, be 'off target effects'. Even if they demonstrate effectiveness in other cell types, this could be discussed.

The bulk of the paper consists of technical validation – showing that contraction is induced as hoped, and can be used in a variety of ways to measure tissue rheology, kinetics of contractile behavior etc. These studies appear to be thorough. In the discussion, for those not in the field, it would be valuable to address how the measurements made possible by the technique can shed light on specific biological questions, going beyond providing precise quantitation of tissue properties.

In the experimental section there is no information about the cell lines beyond the fact that they were provided by another lab. What is the heterogeneity in expression of the

optoRhoA? What sort of expression levels are required to produce contraction? How were these cell lines constructed and characterized? If there is sufficient published information, the reference can be provided.

Finally a minor point - the way the lines of force are shown in Fig. 2B is easy to see. In Fig 1C one must blow up the PDF on a computer screen due to poor contrast with the complex grey background.

Response to the referees

Dear referees,

Thank you for your detailed and positive feedback on our recent submission to Nature Communications, ID NCOMMS-21-50274, " Light-driven biological actuators to probe the rheology of 3D microtissues". You raised several good suggestions for improving the manuscript. We have now revised the manuscript accordingly, and feel that the changes address all of the concerns raised. In particular, we have added the following new data to (i) quantify long term impacts of optogenetic stimulations on tissue mechanics and (ii) demonstrate that our approach allows, by using cells as local mechanical actuators, the mechanical discrimination between healthy and pathological microtissues: - **we quantified the impact of stress amplitude on the measured stiffness E and delay τ_s between maximum stress and ϵ_{xx} strain** (Supp. Fig. 4). We measured no significant differences in elastic modulus, while the delay τ_s increased with the amplitude of contraction, which evidences the stress-dependent dynamic of strain propagation in fibrous microtissues.

- **we demonstrated the independence of the measured mechanical properties on the number of stimulations**, if these stimulations are spaced far enough apart in time to allow complete relaxation (Supp. Fig. 5).
- **we explored the possibility to track changes in tissue mechanics during fibrogenesis**. We thus observed a strong increase of the elastic modulus E and the delay τ_s during TGF β -induced differentiation of fibroblasts into myofibroblasts (Supp. Fig. 6).
- we investigated whether optogenetics could also be used to influence tissue formation and contractility (Supp. Fig. 7). **We thus demonstrated that a long, repetitive optogenetic conditioning during tissue formation (1 stimulation every 5 min for 24 h) induces an increase in tissue contractility** via a higher expression and/or maturation of the actomyosin machinery.

Please find below a point-by-point response to the concerns you raised. All changes in the manuscript are highlighted in yellow.

Reviewer #1:

R1.1. As a weakness, the study does not present major new technological advancements or biological insight into the regulation of tissue mechanics. The overall capacity of the system seems to be not fully exploited and a detailed physical description or modelling approach has not been performed, which limits new physical and biological insight.

We agree with the reviewer that the capacity of our system was not fully exploited. We have now provided data demonstrating that optogenetics can be used to track changes in tissue mechanics during fibrogenesis. To this end, we treated our optogenetic microtissues with TGF- β 1, a strong fibrogenic growth factor known to induce the differentiation of fibroblasts to myofibroblasts in vivo¹, but also in microtissue models²⁻⁴. We first characterized the differentiation of opto-RhoA fibroblasts into opto-RhoA myofibroblasts, by observing a strong increase of α -SMA stress fibers after 3 days of treatment with TGF- β 1. As the contractility of these myofibroblasts remained photo-activatable, we assessed the elastic modulus E and the delay τ_s by optogenetics after 24 h and 72 h of culture with or without TGF- β 1. Similarly to previously reported measurements³⁻⁵, we measured a strong increase of tissue stiffness for TGF- β 1-treated microtissues at day 3. We also observed that the delay between maximum stress and stretch is almost twice as long with versus without TGF- β 1 after 72 h, which supports a stress-dependent dynamic of strain propagation, consistent with the catch bond behavior of the contractile actomyosin machinery^{6,7}.

On a more physical aspect, recent work in the field suggested that the history of the mechanical activity of cells regulates cell spreading, tissue formation and the overall tensional homeostasis of the system⁸⁻¹¹. We used the spatio-temporal control of cell contractility uniquely granted by optogenetics to locally modulate the formation and tensional state of microtissues. To this end, we conditioned the top half of a square microtissue with repetitive stimulations during its formation. Once formed, we interrogated the tissue using optogenetics and observed an increase in contractility of this conditioned area, paralleled with a slightly higher signal of actin fluorescence, which shows that a long, repetitive optogenetic conditioning during tissue formation impacts the expression and/or maturation of the actomyosin machinery. These results open an exciting avenue for simultaneously guiding and interrogating tissue formation, as well as for probing elastic and plastic ECM deformations occurring during morphogenesis¹²⁻¹⁵.

Altogether, thanks to Reviewer #1's suggestion, these additional results highlight the potential of our approach to shed light on specific biological and physical questions about the regulation of tissue mechanics.

Changes in the manuscript:

We added new figures (Supp. Fig. 6, Supp. Fig. 7) presenting (i) the evolution of E and τ_s of microtissues during myofibroblastic differentiation, and (ii) the conditioning of microtissue parts with repetitive localized light-stimulations during tissue formation.

We also added these sentences to the manuscript:

p.12: " Finally, we explored the possibility to track changes in tissue mechanics during fibrogenesis. To this end, we induced the differentiation of fibroblasts to myofibroblasts by incubating microtissues with TGF- β 1, a well-known, strong fibrogenic growth factor^{1,3,5}. We first confirmed the differentiation of opto-RhoA fibroblasts into opto-RhoA myofibroblasts by observing a strong increase of α -SMA stress fibers after 3 days of treatment with TGF- β 1 (Supp. Fig. 6.A-C). We then assessed the elastic modulus E and the delay τ_s by optogenetics after 24 h and 72 h of culture with or without TGF- β 1. Similarly to previously reported measurements³⁻⁵, we observed a strong increase of tissue stiffness for TGF- β 1-treated microtissues at day 3 (Supp. Fig. 6.D). We also observed that the delay τ_s between maximum stress and stretch is almost twice as long with versus without TGF- β 1 after 72 h (Supp. Fig. 6.E), which either supports the actin-myosin catch bond behavior or suggests a different relaxation dynamic of myofibroblasts versus fibroblasts. "

p.14: "Finally, we investigated whether optogenetics could also be used to influence this architecture and the resulting local contractility. We conditioned the top half of a square microtissue with repetitive stimulations every 5 minutes for the first 24 h of its formation, before probing its contractility with a single, narrow, perpendicular pulse (Supp. Fig. 7.A-B). We thus observed a larger compression of the conditioned top half, reaching 3.0 ± 0.5 %, compared to a 2.1 ± 0.7 % compression of the non-conditioned bottom half (Supp. Fig. 7.C-E). The top half of the microtissues also exhibited a slightly higher signal of actin fluorescence (Supp. Fig. 7.F-H) but no difference in collagen fluorescence (Supp. Fig. 7.I-K) after immuno-staining. Altogether, these results suggest that, similarly to long term, external mechanical stimulations, a long term optogenetic conditioning during tissue formation impacts the expression and/or maturation of the actomyosin machinery^{16,17}, with the additional ability to do it locally.

Consequently, the use of optogenetically-modified cells as actuators to probe their environment appears as a powerful tool to guide tissue formation while simultaneously probing non-destructively architectural heterogeneities."

R1.2. The quantitative analysis appears limited and does, for example, not address specific power law or exponential response behaviours associated with previously described viscoelastic tissue models and studies employing external perturbation.

Previous work using a vacuum-actuated stretcher characterized extensively the time-dependent stress relaxation and recovery responses of microtissues over three decades of time, using a stretched exponential model². In our work, we demonstrate the potential of using optogenetic cells as mechanical actuators to probe local tissue mechanics "from the inside" with physiologically relevant stresses. Thus, the possible actuation profiles are imposed by the dynamic of both the optogenetic construct and the RhoA pathway. The optogenetic construct CRY2/CIBN is characterized by a fast recruitment of optoGEF-RhoA to its targeted location (seconds) and a slower dissociation (minutes)^{18–20}, while the dynamic of the RhoA pathway leads to a contractility activation in the order of tens of seconds and a slower relaxation of several minutes¹⁹. The possible stimulation frequencies are thus not compatible with classical mechanical assays that apply step changes in strain of vary the stimulations frequencies over several orders of magnitudes. Nevertheless, we now demonstrate the possibility to describe our results with mathematical models, previously proposed for similar microtissues, and composed of series of springs and dashpots^{2,14,21}. We were able to estimate spring constants and viscosity, depending on the model used to fit the experimental data, thus confirming the ability of our method in giving access to the rheology of microtissues by simply using light stimuli.

Changes in the manuscript:

We added a new supplemental figure (Supp. Fig. 2) presenting the modeling of the results obtained in Fig. 4 by three rheological models: the Kelvin-Voigt (KV) model, the standard linear solid (SLS) model and the stretched exponential model.

We also added these sentences to the manuscript:

p.10:" Of note, although the range of possible frequencies is limited compared to external actuators, the stress-strain curves we obtained could be analyzed with rheological models commonly used for collagenous microtissues (Supp. Fig. 2), such as the Kelvin-Voigt model (a spring and a dashpot in parallel), the standard linear solid (SLS) model (a spring and a dashpot in series, in parallel to another spring), or the stretched exponential model (a sum of Maxwell bodies, i.e. a spring and a dashpot in series, with a specific distribution of time constants)^{2,21}."

R1.3. Strain was generated and assessed within a small range and the authors should clarify if this range can be extended and how a broader strain variation affects measured tissue parameters. Along this line, the authors vary illumination area but not the illumination time. Would an increase in the illumination time lead to a larger contractile stress or did local tissue contraction already saturate for the given settings? How would the variation of illumination time affect measured tissue parameters?

In our initial work, we fixed the number, intensity and duration of the light pulses as we wanted to focus on the rheological aspect of our approach. However, we agree with the reviewer that more details about the range over which we can vary the illumination parameters, and their impact on the measurement, are necessary. In our revised manuscript, we quantified the influence of the number and the energy of stimulations on the tissue contraction and the measured mechanical properties. As the photo-activation of CRY2 depends on both the amplitude and the frequency of the light pulses^{18,20}, we varied the amplitude of contraction by varying either the irradiance of a single light pulse, or the number of 1-minute-spaced light pulses. We measured no significant differences in elastic modulus, while the delay τ_s between maximum stress and ϵ_{xx} strain increased with the amplitude of contraction. Thanks to the reviewer's comment, we thus evidenced (i) the independence of the measured stiffness on the stimulation parameters, and mostly (ii) the stress-dependent dynamic of strain propagation in fibrous microtissues, consistent with a catch bond behavior of the contractile actomyosin machinery^{6,7}.

Changes in the manuscript:

We added a new supplemental figure (Supp. Fig. 4) presenting the impact of stress amplitude on measured stiffness E and delay τ_s , as well as the associated paragraph:

p.11: "We interrogated the dependence of the measured elastic modulus and time delay on the amplitude of the light-induced contraction. As the photo-activation of CRY2 depends on both the amplitude and the frequency of the light pulses^{18,20}, we varied the amplitude of contraction by varying either the irradiance of a single light pulse, or the number of 1-minute-spaced light pulses (Supp. Fig. 4). We measured no significant differences in elastic modulus, while the delay τ_s between maximum stress and ϵ_{xx} strain increased with the amplitude of contraction. [...] Altogether, these results demonstrate that using the cells themselves as mechanical actuators allows to probe, as the cells naturally do, reversibly and non-destructively, the elasticity of fibrous microtissues. Moreover, this approach evidences the stress-dependent dynamic of strain propagation in such tissues, consistent with a catch bond behavior of the contractile actomyosin machinery^{6,7}."

R1.4. Can local tissue areas be repetitively stimulated for contraction and how would this affect tissue mechanical properties? It would also be relevant to address if the use of cells as mechanical actuators can affect measured tissue properties differently compared to externally applied force measurements.

In order to easily perform rheological measurements, we used stimulation parameters inducing an elastic response of the microtissue and avoiding any plastic deformation. We now provide new results demonstrating that (i) successive stimulations, spaced far enough apart in time to allow complete relaxation (30 min between each stimulation), do not affect tissue mechanical properties, whereas (ii) rapid successive stimulations during tissue formation induce local changes in tissue contractility and actin organization (see R1.1).

Changes in the manuscript:

We added two new supplemental figures (Supp. Fig. 5 and Supp. Fig. 7), the first one presenting the independence of the mechanical properties on the number of stimulations, and the second one the mechanical conditioning using repetitive light stimulations during tissue formation (see R1.1).

We also added these sentences to the manuscript (see R1.1 for the paragraph related to Supp. Fig. 7):

p.11: " We further verified that the light-induced contraction was not inducing plastic, irreversible deformation of the collagen and/or actomyosin network. To this end, we illuminated the left half of microtissues with six successive light stimulations, spaced far in time (30 min between each stimulation) to allow complete relaxation. We did not measure any difference in elastic modulus or time delay between the different stimuli (Supp. Fig. 5)."

R1.5. The authors refer to previously published work how differences in cantilever stiffness can affect tissue stiffness and viscosity but this remains speculative if not explicitly tested experimentally.

We understand from the reviewer's comment that the influence of cantilever stiffness on tissue mechanical properties was not clear enough in our manuscript. The dependence between tissue stiffness and the spring constant of the cantilevers was explicitly tested by Zhao et al. using magnetically-actuated cantilevers³. In order to compare and support our measurements, we confirmed these results using our optogenetic approach and we evidenced a stress-dependent dynamic of strain propagation (see R1.1)

Changes in the manuscript:

p.10: " To this end, we stimulated locally microtissues suspended between micropillars presenting a spring constant between 0.20 to 1.12 N/m, which have been previously shown to influence tissue contractility and stiffness^{3,22,23}. We measured the resulting stretch of the non-stimulated zones and inferred elastic moduli and temporal delay between maximum stress and stretch. In good agreement with previous measurements using various spring constants³, we found that the resting stress as well as the elastic modulus increase from 6.2 ± 2.5 to 21.4 ± 8.3 kPa with increasing micropillar spring constant (Supp. Fig. 3)."

R1.6. To extend the biological insight of the current study a specific perturbation of cytoskeletal elements, cell-matrix adhesion or mechanosensitive pathways could be performed to gain insight into the regulation of tissue mechanics beyond previous reports.

We agree with Reviewers #1 and #3 who both suggested to demonstrate how our approach could shed light on specific biological questions. To this end, we performed two new experiments demonstrating (i) the change of mechanical properties induced by the TGF β ₁-induced differentiation of fibroblasts into myofibroblasts and (ii) the impact of rapid successive stimulations during tissue formation on tissue contractility and actin organization (see R1.1).

Changes in the manuscript: We added new figures (Supp. Fig. 6, Supp. Fig. 7) presenting (i) the evolution of E and τ_s of microtissues during myofibroblastic differentiation, and (ii) the conditioning of microtissue parts with repetitive localized light-stimulations during tissue formation, as well as the corresponding paragraphs (see R1.1).

R1.7. The basics of the culture technique should be more clearly described both in the schematic figure 1A and the methods section, in addition to referring to previously published work. It should be emphasized that the microtissue is free-hanging and clearly described how this is achieved, given that this has major relevance for the tension measurements described. We apologize to Reviewer #1 if this point was confusing. We added images and schematics to Figure 1, as well as more detailed information in the Materials and Methods section, to better describe microtissue formation.

Changes in the manuscript:

We added a panel to Fig. 1 showing representative images and schematics that describe the formation of microtissues. We also added these sentences to the manuscript:

p.20: " Over these first 24 h of cultivation, the fibroblasts spread and spontaneously compact the collagen matrix. The two T-shaped cantilevers anchor and constrain the contraction of the collagen matrix to form a microtissue that spans across the top of the pair of cantilevers."

R1.8. Regarding the cell displacement analysis, the authors should make a clearer distinction between cell elongation versus cell displacement/flow and how it would influence tissue visco-elasticity.

We assumed in our work that cells do not migrate over the duration of a contraction, i.e. 20 minutes. Considering an average migration speed of $\sim 0.1 \mu\text{m}/\text{min}$ for NIH 3T3 fibroblasts²⁴, we assumed cell displacement to be negligible compared to cell elongation ($\sim 1 \mu\text{m}/\text{min}$). In similar fibroblast-populated microtissues submitted to axial strain, Walker et al. previously showed that the axial strain produced by cell lengthening and rotation were $\sim 85\%$ and $\sim 15\%$ of the axial tissue strain, respectively, which further supports our assumption¹⁶.

In our displacement analysis, we considered the tissue as homogeneous and we did not discretize it as individual cells and matrix. As such, we measured tissue displacement and corresponding tissue deformation. Although further experiments might be done with other microscopy techniques such as light sheet microscopy, to better characterize the light-induced strain in 3D, our PIV analysis already gives an easy access to strain maps with a $25 \mu\text{m}$ resolution without requiring complex imaging systems, thus making it accessible to most laboratories.

Changes in the manuscript:

We added the following sentences p. 22: "The tissue was considered as homogeneous and was not discretized as individual cells and matrix. Consequently, we measured tissue displacements and corresponding tissue deformations. Of note, cell migration over the duration of a contraction was considered negligible compared to cell deformation."

R1.9. Analysis of actin/collagen alignment: How do the authors distinguish actin alignment from cell shape? Cortical actin appears very bright and cells are elongated along the x-direction. This could yield a cell alignment parameter rather than an actin fibre alignment. The resolution in Fig 2J further appears limited to analyse collagen alignment. The chosen colour bar for actin/collagen alignment also makes it difficult to distinguish the alignment in parallel and vertical direction (pink= 0° and red 90° are similar). Also, what does the white colour indicate?

Cortical actin is indeed prominent and thus correlates with cell shape. We have added this precision to the manuscript. We understand from the reviewer's comment that our choice of representation of the actin and collagen alignment were confusing. We modified Fig. 2 to clarify this point. We now present a magnification of a representative confocal slice for actin and collagen, with the corresponding orientation maps, and we changed the color scale to facilitate reading.

Changes in the manuscript:

We modified panels G, H, J and K in Fig. 2, to show representative confocal slices for actin and collagen, with the corresponding orientation maps, and we changed the color scale to facilitate reading.

We also added these sentences to the manuscript:

p.6: " We thus found that the longitudinal contraction induced by light correlated with a strongly anisotropic architecture of both actin and collagen, with more than 80 % of the actin fibers and more than 70 % of the collagen fibers aligned within less than 30° of the x-axis (Fig. 2.G-L, Supp. movie 3). As cortical actin is prominent, cell alignment correlates with actin alignment, which is coherent with the fact that fibroblasts line up and remodel their extracellular matrix to align with the principal maximum strains developed during tissue formation^{9,25,26}."

R1.10. Figure 2: The authors argue that the anisotropic orientation of actin and collagen fibres are responsible for the anisotropic propagation of mechanical stimulation from an isotropic stimulus. How do the authors argue about the fact that the applied stimulus itself is not isotropic? (as it's not equidistant from the edges the force experienced by the different regions of the tissue will not be symmetric as in Figure 5)

We thank the reviewer for pointing out this confusing statement. The stimulus is indeed not equidistant from the edges. However, as the diameter of the stimulus is approximately a third of the tissue width and a fifth of the tissue length, the ϵ_{yy} strain should be larger than the ϵ_{xx} strain if the microtissue were isotropic. In order to avoid any confusion, we deleted the isotropic qualification of the stimulus in this section.

R1.11 Figure 3: It is not fully clear how stress (σ_{xx}) was calculated and a more detailed description in the method section would be beneficial ("deduced from the pillar deflection and the cross-section of the microtissue". Authors use ϵ_{xx} and "strain" interchangeably within the same figure, it would be easier to understand if this labeling was consistent

The stress σ_{xx} was calculated by dividing the force, inferred from the pillar deflection, by the cross-sectional area in the center of the tissue, estimated from confocal z-stack images. We now specify this point and we harmonized the labeling of the strains.

Changes in the manuscript:

We checked the consistency of labeling for strains and we added these sentences to the manuscript: p.21: " The stress σ_{xx} was calculated by dividing the force by the cross-sectional area measured in the center of the tissue, estimated from confocal z-stack images ¹⁷."

R1.12. Figure 5: It is not clear whether collagen is present in the center of Figure 5O and the related supplementary movie 8, can the authors comment?

There is collagen in the center of the tissue, although less and less organized than along the sides of the tissue. We agree with the reviewer that this point is not clear and we modified the panels M and P in Fig. 5 to present magnified representative confocal slices for actin and collagen, with the corresponding orientation maps, and we changed the color scale to facilitate reading (see R1.9).

Changes in the manuscript:

We modified panels M and P in Fig. 5. We now present magnified representative confocal slices for actin and collagen, in the center and on the side, with the corresponding orientation maps.

R1.13. Can the authors also clarify the reasoning of this sentence "In order to infer the rheology of the microtissue, we assimilated the stimulated part of the tissue to an actuator stretching the non-stimulated part."

This sentence describes the concept of our work, i.e. using the contraction of one part of the tissue (the stimulated one) to stretch the other part (the non-stimulated one), thus probing the mechanical properties of the latter. We tried to clarify this point by modifying this sentence.

Changes in the manuscript:

p. 9: We replace the quoted sentence by the following "In order to probe the mechanical properties of the microtissue, we used the stimulated half of the tissue as a mechanical actuator stretching the non-stimulated half."

R1.14. The authors could also more extensively discuss limitations of their presented approach, such as the need of keeping the actomyosin cytoskeleton intact to use cells as internal actuators, which complicates the study of the role of actin/myosin or actin binding proteins in regulating tissue mechanics (and a significant role has been attributed to the actomyosin cytoskeleton in previous work).

We agree with the reviewers that the limitations of our approach were not discussed enough. We now modified the manuscript to include this discussion.

Changes in the manuscript:

We added the following sentences p. 17: "As our approach uses the contraction of one part of the tissue (the stimulated one) to stretch the other part (the non-stimulated one) to probe the mechanical properties of the latter, it inherently requires an active actomyosin machinery, which could complicate the study of the role of cytoskeletal components in regulating tissue mechanics. Indeed, a complete inhibition of the contractile ability of the microtissue, using high doses of actomyosin-targeting drugs for example, would hinder the use of our approach. However, our method remains valid as long as the contractile machinery is at least partially active, e.g. for small to medium doses of actomyosin-targeting drugs."

Reviewer #2:

Summary: The authors present an interesting application of optogenetics to assess the mechanical behavior of microtissues through light-sensitive cell-driven contractions. The ability to characterize a tissue using intrinsic actuators could be quite valuable for both characterization of microtissues and better understanding the mechanisms of cellular behavior.

The study demonstrates the value of this method for assessing microtissue mechanical behaviors and studying how various factors can influence the structure and properties of the microtissues. The study is, however, limited in its verification of this method for measuring mechanical properties and incomplete in its assessment of the anisotropy of the microtissues.

R2.2. While the obtained elastic modulus is said to be in “very good agreement” with the results of prior studies, I note that different parameters were used in those studies. This study would be strengthened by performing a more direct verification, i.e. assessing the mechanical properties of a set of samples using both light-stimulated contractions and some other means (such as the magnetically actuated microcantilevers). This verification is very important for this manuscript, since the manuscript is focused on presentation of a method. Additionally, some verification of the viscosity measurements, either through references or preferably direct comparisons should be provided.

We performed a new set of experiments using the same parameters than Zhao et al. who used magnetic actuation to probe the stiffness of fibrous microtissues and we found very similar results. By looking further into the apparent viscosity of microtissues, we found that the delay τ_s between maximum stress and ϵ_{xx} strain was not due to a passive viscosity but rather to a stress-dependent dynamic of strain propagation, consistent with a catch bond behavior of the contractile actomyosin machinery^{6,7}.

Changes in the manuscript:

We modified Supp. Fig. 3 to include a comparison with stiffness data previously obtained with the same parameters and we modified the discussion about tissue viscosity according to our new findings presented in Supp. Fig. 4 and Supp. Fig. 6 (see R1.3).

R2.3. Assessment of anisotropic behavior should more clearly delineate the different forms of anisotropy at play in this context (anisotropy of mechanical properties, anisotropy of cellular contractions, and anisotropy of fiber orientations). Figure 2 shows that strains in the x-direction are significantly greater in magnitude than strains in the y-direction, this behavior could be attributed to anisotropy of the elastic modulus and/or anisotropy of the cellular contractions, both of which could be related to the anisotropic orientation of actin and collagen. The two-cantilever set up, which is only capable of determining x-direction forces, thus seems ill-suited to fully analyze the anisotropy of the tissue.

We agree with the reviewer that the two-cantilever set up does not allow to delineate the different forms of anisotropy. However, the four-cantilever set up allowed us to establish that the light-induced strain orientations correlated with the anisotropy of fiber orientations (Fig. 5). By performing a discoidal, isotropic light stimulation either in the center or on the left side of these square microtissues, we observed a corresponding either isotropic or anisotropic strain field, that correlate with mostly random or more anisotropic organization of cytoskeletal and extracellular architecture, respectively. Bose et al. previously showed, using a computational modeling approach, that aligned collagen regions of microtissues were six times stiffer than unaligned regions⁹. Altogether, these results demonstrate that our optogenetic approach, combined with specifically designed microtissue geometry and computational modeling, could offer an interesting approach to analyze the complex interplay between mechanical boundary constraints, cell contractility, ECM density, alignment, and mechanical properties.

Changes in the manuscript:

We added the following sentences p. 18: " Altogether, our work thus demonstrates that an optogenetic control of cell contractility, combined with specifically designed microtissue geometry and possibly computational modeling⁹, could offer a powerful approach to analyze the complex interplay between mechanical boundary constraints, cell contractility, ECM density, alignment, and mechanical properties."

R2.4. The strain fields shown (e.g. Figure 2D and 2E) are astonishingly smooth. In this reviewer's experience, which is backed up by work by others (e.g. Midgett et al. Acta Biomater. 2017 Apr 15;53:123-139. doi: 10.1016/j.actbio.2016.12.054. Epub 2017 Jan 17), simply differentiating the displacement field numerically, as indicated in the section "Force measurement and strain measurement" results in noisy strain fields. Please provide more information on the method used, beyond simply citing reference 36, as the type of noise present in the images and displacement field depend strongly on the mode of imaging, contrast, and resolution.

We understand from the reviewer #2's comment that the methods used for strain measurement were not described precisely enough and we modified accordingly the corresponding paragraph in the Materials and Methods section. Briefly, analyzed images were 1024x512 pixels (800x400 μm) or 1024x1024 pixels (800x800 μm) for microtissues between 2 or 4 pillars, respectively. Data were analyzed in three successive passes. The first pass using relatively large interrogation areas of 128x128 pixels (100x100 μm) to calculate the displacement reliably while minimizing the signal-to-noise ratio. The second and third passes used successively decreasing interrogation areas of 96x96 and 64x64 pixels (corresponding to 75x75 and 50x50 μm , respectively), as well as the displacement information obtained during the previous pass as an offset. For each pass, the overlap between interrogation areas was set to 50%, leading to a final resolution of 25 μm . Finally, outliers were filtered using a local normalized median filter²⁷, missing vectors were replaced by interpolated data²⁸ and the noise was reduced using a penalized least squares method²⁹.

The "smoothness" of our results can be explained by 2 main factors:

- Fibrous microtissues composed of fibroblasts and collagen are more homogeneous than native explants such as lamina cribrosa presented in Midgett et al. ³⁰, and the strain fields generated in such microtissues by others using external stimulations are smoother ^{16,17,31}.
- As stated in the caption, the strain fields presented in Fig. 2D and 2E are obtained by averaging the strain fields of 25 microtissues submitted to the same stimulation, which induces an apparent smoothness of the field. Individual, more "noisy" strain fields are presented in the supp. movies 2, 4, 5, 6, 7.

Changes in the manuscript:

We added the following sentences to the Materials & Methods section:

p.21; "Analyzed images were 1024x512 pixels (800 x 400 μ m) or 1024x1024 pixels (800x800 μ m) or 1024x1024 pixels for microtissues between 2 or 4 pillars, respectively, and the size of the interrogation areas was successively reduced from 128x128 to 96x96 and finally 64x64 pixels (corresponding to 100x100, 75x75 and 25x25 μ m, respectively). For each pass, i.e. for each size of interrogation area, the overlap between interrogation areas was set to 50%, leading to a final resolution of 25 μ m. Outliers were filtered using a local normalized median median filter ²⁷, missing vectors were replaced by interpolated data ²⁸ and the noise was reduced using a penalized least squares method ²⁹."

R2.5. Figure 3E appears to show higher stiffness in tension vs. compression. This is an interesting result in regards to the purported interest in comparing cell-generated forces to tissue stiffness. Some discussion of this finding would strengthen the manuscript.

We thank the reviewer for this interesting point of discussion. To our knowledge, the mechanical properties of collagenous tissues such as ours have never been fully characterized in compression. However, previous work has shown the highly nonlinear behavior of collagen gels, with a lower stiffness in compression compared to stretch ^{32,33}. Currently, we cannot extract mechanical properties from the compression of the stimulated tissue area without an appropriate modeling of our system but we now discuss this point in the revised manuscript.

Changes in the manuscript:

We added the following sentences to the discussion section:

p. 16: " We thus used optogenetic cells as biological actuators to probe the mechanics of fibroblasts/collagen microtissues. By tracking the maximum amplitudes of stress, compression and stretch in response to local light-activation of the contractility, we were able to plot hysteretic stress/strain curves, as the strains were delayed in time compared to the stress. We first observed that these fibrous microtissues seemed more compliant in compression than in stretch, as the compressed, stimulated part was more deformed than the stretched, non-stimulated one. Previous work has shown the highly nonlinear behavior of collagen gels, with a lower stiffness in compression compared to stretch ^{32,33}, and an appropriate modeling of our light-sensitive microtissues would be particularly relevant to characterize this mechanical behavior in more details."

R2.6. In the 'Inferring mechanical properties of stretched areas' section, the authors report that "force increase was proportional to the area of the stimulated zones" and from that infer that light-induced contraction is proportional to the number of stimulated cells, but there is no accounting for the potential influence of cell density. Would a higher cell density and a smaller area of stimulation cause the same force increase as a proportionally lower density and larger area?

In theory, this experiment could provide interesting information about the relationship between tissue contractility and cell density. However, it is currently impossible to address it with our system, as cell density strongly affects tissue formation, organization and contractility ^{21,22} (see also figure below) which, ultimately, strongly influences the light-induced force increase.

Figure 1. Influence of the cell density on tissue formation. Temporal evolution of microtissues composed of low and high cell densities. Scale bar is 100 μ m.

R2.7. Given that part of the focus of the measurements is on time-dependence of the contraction and relaxation, I suggest the study also include multiple cycles of contraction. This would provide insight into whether the contractions are inducing more permanent matrix changes, or at least matrix changes that accumulate with repeated loading, whether this changes the viscosity, and how this affects the ability of the cells to continue inducing tissue contraction. Our work aims at demonstrating the possibility of using cells as mechanical actuators to probe tissue material properties. To this end, we used stimulation parameters inducing an elastic response of the microtissue and avoiding any plastic deformation. However, we understand from Reviewers #1 and #2 that a better quantification of this long term impact is important (see also R1.1 and R1.4). We now provide new results demonstrating that (i) successive stimulations, spaced far enough apart in time to allow complete relaxation, do not affect tissue mechanical properties, whereas (ii) rapid successive stimulations during tissue formation induce changes in tissue contractility and actin organization.

Altogether, thanks to Reviewers #1 and #2's suggestions, these additional results highlight the potential of our approach for simultaneously guiding and interrogating tissue formation, as well as for probing elastic and plastic ECM deformations occurring during morphogenesis^{12–15}.

Changes in the manuscript:

We added two new supplemental figures (Supp. Fig. 5 and Supp. Fig. 7), the first one presenting the independence of the mechanical properties on the number of stimulations, and the second one the mechanical conditioning using repetitive light stimulations during tissue formation.

We also discussed these results p. 11 and p. 14 (see R1.1 and R1.4).

R2.8. Introduction, 2nd paragraph, “it remains problematic to assess how such external mechanical stimulations compare with internal, cell-generated contractions”: This argument would be stronger

if the manuscript provided a rationale for making this comparison. What is the biological implication?

Cells are constantly pulling and pushing on their microenvironment, thus assessing, among other things, its mechanical properties³. However, this environment is often a heterogeneous, non-linearly elastic network of fibers, and the characterization of its mechanical properties is thus very sensitive to the probe size and the loading history of the mechanical test device used for such characterization. The concept of our work consists in using the cells themselves as mechanical actuators, in order to probe the mechanical properties of the tissue "from the inside", as the cells naturally do it.

Changes in the manuscript:

We emphasized this rationale in the introduction p. 2:

"A key limitation to the study of mechanical signals in tissue biology is to assess how external mechanical stimulations such as magnetic or vacuum actuations compare with internal, cell-generated contractions. Indeed, cells are constantly pulling and pushing on their microenvironment, thus assessing, among other things, its mechanical properties³. As rheological measurements are very sensitive to the probe size and the loading history of the mechanical test device used for such characterization^{35,36}, probing the mechanical properties of the tissue "from the inside", as the cells naturally do it, would further our understanding of how cells investigate their environment and how cell-generated mechanical signals propagate. However, no method currently exists that can locally modulate the contractility of a group of cells within a 3D tissue while simultaneously measuring global tissue contractility as well as both fine-scale cytoskeletal and extracellular architecture. "

R2.9. Pg. 6, "This anisotropic contraction was paralleled with a strongly anisotropic field of deformation": Force was only measured in one direction, so there is no way to know from the data collected if the forces are anisotropic; instead, we know only that the contractile deformation is anisotropic, which is the same thing as the deformation field being anisotropic. Thus, the two quantities described as "parallel" to each other are one and the same.

We thank the reviewer for pointing this redundancy. We corrected the text accordingly.

Changes in the manuscript:

p.6: We replaced the initial sentence with "This anisotropic contraction corresponds to a strongly anisotropic field of deformation, [...]"

R2.10. Pg. 11 – A doubling of the elastic modulus from 15.1 kPa to 28.5 kPa from 24-48 hours is reported, however on the previous page, an average elastic modulus of 19.2 kPa is reported. Since the 19.2 kPa measurement was presumably taken at 24 hrs, it's not clear why there's a difference between the 15.1 kPa and 19.2 kPa measurements.

The experimental conditions in both experiments are indeed the same. The discrepancy between the reported values of 15.1 ± 6.5 kPa (over 60 microtissues) and 19.2 ± 5.0 kPa (over 20 microtissues) come from the variability between two sets of experiments and is represented by the standard deviations.

R2.11. The figures clearly show that the cross-sectional area of the tissue is not constant, so some explanation of how this was accounted for in the calculation of stress is needed.

The cross-sectional area was measured in the center of the microtissues, similarly to previous work^{3,17,22,23}. We now give more details about this parameter in the Materials and Methods section.

Changes in the manuscript:

p.21: " The stress σ_{xx} was calculated by dividing the force by the cross-sectional area measured in the center of the tissue, estimated from confocal z-stack images."

R2.12. Discussion, "whereas collagen density or integrity only impact tissue stiffness": no measurement of collagen integrity is provided.

We understand this sentence was confusing, as we meant that the degradation of collagen by the enzyme collagenase impacted tissue stiffness. We corrected this sentence in the manuscript.

Changes in the manuscript:

p.16: We replaced the initial sentence with " Using this optogenetic approach, we found that boundary conditions, i.e. pillar rigidity, affect the stiffness of microtissues and, because of their impact on cell-generated stress^{3,22}, the delay between stress and strain, whereas collagen density or its degradation by collagenase only impact tissue stiffness. "

Reviewer #3:

I believe the authors are correct in saying that optogenetics has not been used to control specific cells within a population and thereby control force in the tissue. The work is therefore significant and presents a potentially useful new tool. However, I believe there are some issues that need to be addressed:

R3.1. The method that was actually used to induce cell contraction is published, so the novelty of the paper does not lie in developing this method, but in applying it in a new way. I believe there may be issues with the optoRhoA method as used here. Opto-RhoA functions by recruiting a Rho GEF to the membrane. Does this necessarily lead only to contraction? It is possible that the RhoGEF is completely specific for RhoA, but it would be valuable to have a discussion of the level to which this has been proven. More importantly, Rho activates and inhibits multiple downstream pathways, not just contraction. Formin activation in fact leads to protrusion. The experiments in the paper show that, regardless of what other effects are induced in this specific cell type and tissue, contraction is produced. Is this true in other cells and circumstances? Is there complex biology with some cells protruding and others contracting (e.g. cell versus edge of irradiated region), with net contraction. Indeed, RhoA has multiple downstream effectors. Previous work using 3T3 fibroblasts or MDCK epithelial cells demonstrated that over time, contractions induced by long-lasting (10-60 minutes) light stimulations of RhoA was paralleled by the formation of actin stress fibers, which could induce cell migration²⁴, as well as an increase in nuclear YAP, and that both phenomena were fully reversible¹⁹. In our work, we used single, short stimulations that induced an elastic response of the microtissue, i.e. the tissue comes back to its resting state after contraction, which suggests no long lasting effects such as cytoskeleton remodeling for example. Although not published yet, we also observed similar elastic responses in single opto-fibroblasts³⁷.

However, we cannot exclude the suggestion of the reviewer that there could be "complex biology with some cells protruding and others contracting, with net contraction". Altogether, we understand from the reviewer's comments that a better quantification of the possible long term impacts of optogenetic stimulations is important (see also R1.1, R1.4 & R2.7). We now provide new results demonstrating that (i) successive stimulations, spaced far enough apart in time to allow complete relaxation, do not affect tissue mechanical properties, whereas (ii) rapid successive stimulations during tissue formation induce changes in tissue contractility and actin organization. Altogether, thanks to the reviewers' comments, these additional results demonstrate the potential of our approach for controlling the formation of tissues while simultaneously assessing their mechanical properties and architectural heterogeneities.

Changes in the manuscript:

We added two new supplemental figures (Supp. Fig. 5 and Supp. Fig. 7), the first one presenting the independence of the mechanical properties on the number of stimulations, and the second one the mechanical conditioning using repetitive light stimulations during tissue formation.

We also discussed these results p. 11 and p. 14 (see R1.4).

R3.2. The authors have shown that in the right situation the technique can be useful, but I believe they should discuss caveats of the method, be 'off target effects'. Even if they demonstrate effectiveness in other cell types, this could be discussed.

We agree with the reviewer that we did not discuss enough the limitations of our approach. The CRY2/CIBN optogenetic system is a tool of choice for manipulating protein distribution on the plasma membrane over a wide dynamic range, using a simple procedure for its spatiotemporal control¹⁸. By fusing CRY2 with the catalytic domain of a specific GEF for the Rho GTPase RhoA, Valon et al. thus engineered a tool, using standard transient transfection methods or viral infections to create stable cell lines, that enables to reversibly, locally increase cell contractility in different cell lines^{19,24}. However, this approach presents several limitations that we now discuss in the revised manuscript:

- The optogenetic construct CRY2/CIBN is characterized by a fast recruitment of optoGEF-RhoA to its targeted location (seconds) and a slower dissociation (minutes)¹⁸⁻²⁰, while the dynamic of the RhoA pathway leads to a contractility activation in the order of tens of seconds and a slower relaxation of several minutes¹⁹. As a result, the induced cell contraction and relaxation takes almost 20 minutes, thus precluding its use for probing rapid mechanical changes or quantifying time-dependent mechanical responses of microtissues at different frequencies.
- As discussed previously (R3.1), RhoA has many downstream effectors. Moreover, Dehmelt, Nalbant and coauthors recently demonstrated that Rho GTPase-based signaling networks can generate pulses and propagating waves of cell contractions, via the self-amplification and self-inhibition of RhoA. It is thus not excluded that off target effects could exist, depending on the initial mechano-chemical state of the microtissue and the parameters of stimulation.

Changes in the manuscript:

p.17: We added the following sentences; " However, as our approach uses the contraction of one part of the tissue (the stimulated one) to stretch the other part (the non-stimulated one) to probe the mechanical properties of the latter, it inherently requires an active actomyosin machinery, which could complicate the study of the role of cytoskeletal components in regulating tissue mechanics. Indeed, a complete inhibition of the contractile ability of the microtissue, using high doses of actomyosin-targeting drugs for example, would hinder the use of our approach. Yet, our method remains valid as long as the contractile machinery is at least partially active, e.g. for small to medium doses of actomyosin-targeting drugs. Similarly, although the recruitment of CRY2 to the membrane is very fast (few seconds), its dissociation is slower (several minutes)¹⁸, while the dynamic of the RhoA pathway leads to a contractility activation in the order of tens of seconds and a slower relaxation of several minutes¹⁹. As a result, the induced cell contraction and relaxation takes almost 20 minutes, thus precluding its use for probing rapid mechanical changes or quantifying time-dependent mechanical responses of microtissues at frequencies. Finally, our optogenetic target RhoA has many downstream effectors. Although we used single, short stimulations that induced an elastic response of the microtissue with no long lasting effects such as cytoskeleton remodeling for example, we cannot exclude that off target effects could exist, via for example the self-amplification and self-inhibition of RhoA³⁸. "

R3.3. The bulk of the paper consists of technical validation – showing that contraction is induced as hoped, and can be used in a variety of ways to measure tissue rheology, kinetics of contractile behavior etc. These studies appear to be thorough. In the discussion, for those not in the field, it

would be valuable to address how the measurements made possible by the technique can shed light on specific biological questions, going beyond providing precise quantitation of tissue properties. We agree with Reviewers #1 and #3 who both suggested to demonstrate how our approach could shed light on specific biological questions. To this end, we performed two new experiments demonstrating (i) the change of mechanical properties induced by the TGF β ₁-induced differentiation of fibroblasts into myofibroblasts and (ii) the impact of rapid successive stimulations during tissue formation on tissue contractility and actin organization (see R1.1).

Changes in the manuscript: We added new figures (Supp. Fig. 6, Supp. Fig. 7) presenting (i) the evolution of E and τ_s of microtissues during myofibroblastic differentiation, and (ii) the conditioning of microtissue parts with repetitive localized light-stimulations during tissue formation, as well as the corresponding paragraphs (see R1.1).

R3.4. In the experimental section there is no information about the cell lines beyond the fact that they were provided by another lab. What is the heterogeneity in expression of the optoRhoA? What sort of expression levels are required to produce contraction? How were these cell lines constructed and characterized? If there is sufficient published information, the reference can be provided.

We apologize to the reviewer for not giving such details in our manuscript.

The cell line was previously constructed and used by our collaborators^{18,24}. The DHPH domain of ARHGEF11 Guanine Exchange Factor gene was cloned into CRY2PHRmCherry using Nhe1 and Xho1 cloning sites. ArhGEF11-CRY2PHR-mCherry and CIBN-GFP-CAAX were then inserted into lentiviral backbones (pHR and pLVX respectively) to create the stable cell line of opto-RhoA fibroblasts. After viral transduction, only the top 2% of the cells presenting the highest expression levels for both transgenes were FACS-sorted before amplification. The population is thus not fully but highly homogeneous. We added this information to the Materials and Methods section of the revised manuscript.

Changes in the manuscript: We added the following paragraph in the Materials and Methods section: p.18: "The DHPH domain of ARHGEF11 Guanine Exchange Factor gene was cloned into CRY2PHRmCherry using Nhe1 and Xho1 cloning sites. ArhGEF11-CRY2PHR-mCherry and CIBN-GFP-CAAX were then inserted into lentiviral backbones (pHR and pLVX respectively) to create the stable cell line of opto-RhoA fibroblasts. After viral transduction, only the top 2% of the cells presenting the highest expression levels for both transgenes were FACS-sorted before amplification."

R3.5. Finally a minor point - the way the lines of force are shown in Fig. 2B is easy to see. In Fig 1C one must blow up the PDF on a computer screen due to poor contrast with the complex grey background.

We thank the reviewer for pointing this issue. We now present an outline of the tissue in Fig. 1.D, which makes for a better contrast.

References:

1. Meng, X. M., Nikolic-Paterson, D. J. & Lan, H. Y. TGF- β : The master regulator of fibrosis. *Nat. Rev. Nephrol.* **12**, 325–338 (2016).
2. Walker, M., Godin, M., Harden, J. L. & Pelling, A. E. Time dependent stress relaxation and recovery in mechanically strained 3D microtissues. *APL Bioeng.* **4**, 036107 (2020).
3. Zhao, R., Chen, C. S. & Reich, D. H. Force-driven evolution of mesoscale structure in engineered 3D microtissues and the modulation of tissue stiffening. *Biomaterials* **35**, 5056–5064 (2014).
4. Asmani, M. *et al.* Fibrotic microtissue array to predict anti-fibrosis drug efficacy. *Nat. Commun.* **9**, 1–12 (2018).
5. Walker, M., Godin, M. & Pelling, A. E. Mechanical stretch sustains myofibroblast phenotype and function in microtissues through latent TGF- β 1 activation. *Integr. Biol. (Camb).* **12**, 199–210 (2020).
6. Vernerey, F. J. & Akalp, U. Role of catch bonds in actomyosin mechanics and cell mechanosensitivity. *Phys. Rev. E* **94**, 012403 (2016).
7. Guo, B. & Guilford, W. H. Mechanics of actomyosin bonds in different nucleotide states are tuned to muscle contraction. *Proc. Natl. Acad. Sci. U. S. A.* **103**, 9844–9849 (2006).
8. Kassianidou, E. *et al.* Extracellular Matrix Geometry and Initial Adhesive Position Determine Stress Fiber Network Organization during Cell Spreading. *Cell Rep.* **27**, 1897-1909.e4 (2019).
9. Bose, P., Eyckmans, J., Nguyen, T. D., Chen, C. S. & Reich, D. H. Effects of Geometry on the Mechanics and Alignment of Three-Dimensional Engineered Microtissues. *ACS Biomater. Sci. Eng.* **5**, 3843–3855 (2019).
10. Vignaud, T. *et al.* Stress fibres are embedded in a contractile cortical network. *Nat. Mater.* (2020). doi:10.1038/s41563-020-00825-z
11. Mailand, E. *et al.* Tissue engineering with mechanically induced solid-fluid transitions. *Adv. Mater.* **2106149**, 2106149 (2021).
12. Han, W. *et al.* Oriented collagen fibers direct tumor cell intravasation. *Proc. Natl. Acad. Sci. U. S. A.* **113**, 11208–11213 (2016).
13. Ban, E. *et al.* Mechanisms of Plastic Deformation in Collagen Networks Induced by Cellular Forces. *Biophys. J.* **114**, 450–461 (2018).
14. Liu, A. S. *et al.* Matrix viscoplasticity and its shielding by active mechanics in microtissue models: experiments and mathematical modeling. *Sci. Rep.* **6**, 33919 (2016).
15. Buchmann, B. *et al.* Mechanical plasticity of collagen directs branch elongation in human mammary gland organoids. *Nat. Commun.* **12**, 2759 (2021).
16. Walker, M., Godin, M. & Pelling, A. E. A vacuum-actuated microtissue stretcher for long-term exposure to oscillatory strain within a 3D matrix. *Biomed. Microdevices* **20**, (2018).
17. Zhao, R., Boudou, T., Wang, W.-G., Chen, C. S. & Reich, D. H. Decoupling Cell and Matrix Mechanics in Engineered Microtissues Using Magnetically Actuated Microcantilevers. *Adv. Mater.* **25**, 1699–1705 (2013).
18. Valon, L. *et al.* Predictive Spatiotemporal Manipulation of Signaling Perturbations Using Optogenetics. *Biophys. J.* **109**, 1785–1797 (2015).
19. Valon, L., Marín-Llauradó, A., Wyatt, T., Charras, G. & Trepat, X. Optogenetic control of cellular forces and mechanotransduction. *Nat. Commun.* **8**, 14396 (2017).
20. Kennedy, M. J. *et al.* Rapid blue-light-mediated induction of protein interactions in living cells. *Nat. Methods* **7**, 973–975 (2010).
21. Wakatsuki, T., Kolodney, M. S., Zahalak, G. I. & Elson, E. L. Cell mechanics studied by a reconstituted model tissue. *Biophys. J.* **79**, 2353–2368 (2000).
22. Legant, W. R. *et al.* Microfabricated tissue gauges to measure and manipulate forces from 3D microtissues. *Proc. Natl. Acad. Sci. USA* **106**, 10097–10102 (2009).
23. Boudou, T. *et al.* A Microfabricated Platform to Measure and Manipulate the Mechanics of Engineered Cardiac Microtissues. *Tissue Eng. Part A* **18**, 910–919 (2012).
24. Hennig, K. *et al.* Stick-slip dynamics of cell adhesion triggers spontaneous symmetry breaking and directional migration of mesenchymal cells on one-dimensional lines. *Sci. Adv.* **6**, eaa5670 (2020).
25. Eastwood, M., Mudera, V. C., McGrouther, D. A. & Brown, R. A. Effect of precise mechanical loading on fibroblast populated collagen lattices: Morphological changes. *Cell Motil. Cytoskeleton* **40**, 13–21 (1998).
26. Sander, E. A., Barocas, V. H. & Tranquillo, R. T. Initial fiber alignment pattern alters extracellular matrix synthesis in fibroblast-populated fibrin gel cruciforms and correlates with predicted tension. *Ann. Biomed. Eng.* **39**, 714–729 (2011).

0. Westerweel, J. & Scarano, F. Universal outlier detection for PIV data. *Exp. Fluids* **39**, 1096–1100 (2005).
1. Nogueira, J., Lecuona, A. & Rodriguez, P. A. Data validation, false vectors correction and derived magnitudes calculation on PIV data. *Meas. Sci. Technol.* **8**, 1493–1501 (1997).
2. Garcia, D. Robust smoothing of gridded data in one and higher dimensions with missing values. *Comput. Stat. Data Anal.* **54**, 1167–1178 (2010).
3. Midgett, D. E. *et al.* The pressure-induced deformation response of the human lamina cribrosa: Analysis of regional variations. *Acta Biomater.* **53**, 123–139 (2017).
4. West, A. R. *et al.* Development and characterization of a 3D multicell microtissue culture model of airway smooth muscle. *Am. J. Physiol. Cell. Mol. Physiol.* **304**, L4–L16 (2013).
5. Steinwachs, J. *et al.* Three-dimensional force microscopy of cells in biopolymer networks. *Nat. Methods* **13**, 171–176 (2016).
6. Stylianopoulos, T. & Barocas, V. H. Volume-averaging theory for the study of the mechanics of collagen networks. *Comput. Methods Appl. Mech. Eng.* **196**, 2981–2990 (2007).
7. Eyckmans, J., Boudou, T., Yu, X. & Chen, C. S. A hitchhiker’s guide to mechanobiology. *Dev. Biol.* **21**, 35–47 (2011).
8. Wu, P.-H. *et al.* A comparison of methods to assess cell mechanical properties. *Nat. Methods* **1** (2018). doi:10.1038/s41592-018-0015-1
9. Licup, A. J. *et al.* Stress controls the mechanics of collagen networks. *Proc. Natl. Acad. Sci.* **112**, 9573–9578 (2015).
10. Andersen, T. *et al.* Cell size and actin architecture determine force generation in optogenetically activated adherent cells. *bioRxiv* (2022).
11. Kamps, D. *et al.* Optogenetic Tuning Reveals Rho Amplification-Dependent Dynamics of a Cell Contraction Signal Network. *Cell Rep.* **33**, 108467 (2020).

REVIEWER COMMENTS

Reviewer #1 (Remarks to the Author):

The authors have included several modifications in their manuscript and presented a detailed response to the raised questions. The paper keeps a technical focus and now includes validations of previously measured tissue parameters. New biological insight remains to be obtained with this method. As this is a major limitation of the current work the authors should consider to further strengthen in their manuscript which potential advantages their approach has over other previously used techniques.

Moreover, as the work by Mery et al. aims to establish a methodological advancement over previously established techniques to measure tissue mechanics the authors should further extend their methods section and technical description which remains very concise and mostly refers to previous work. This would make the work more transparent and reproducible for future applications which is not clearly doable at this stage.

Several points regarding the representation of data and conclusion drawn in the manuscript should also be considered by the authors as indicated below.

Related to Fig1 and the assay preparation, the authors revised panel 1A-B but should also include fluorescence microscopy images to show transversal/lateral views of microtissues and their attachment to cantilevers. It remains unclear from the method section how the contraction of tissues leads to their attachment to the cantilever tips. As the deflection of cantilevers is used to derive tissue tension, possible variations in tissue attachments to the cantilevers or friction with the bottom surface could alter measurements or induce high variability and the authors should clarify and clearly describe the methods in the manuscript. Detailed protocols on the methods including for example cell seeding density, quality criteria to assess proper microtissue formation and attachment etc should also be provided.

Were cantilever spring constants measured or calculated? How much was the cantilever deflection after 24h for wild type cells versus the 2% of cells selected for overexpressing the optogenetic construct to assess baseline tissue contraction in light vs dark conditions and the influence of the overexpression of the optogenetic construct.

Fig 1E, the authors should connect tension measured in this panel vs Fig 3C. The stress derivation based on area measures should be exemplified. How and when is area change measured? And how is the area measure obtained in asymmetric vs central stimulation when the optogenetic activation of contraction can be expected to lead to changes in area?

Fig 2G-L and Fig5L-P, the imaging quality and resolution to assess actin and collagen alignment remain limited. Also cell density appears more variable in these panels as for example in Suppl.Fig1 and it appears cells are rounder in the middle region of the tissue versus the margin but a spatial map and how it links to tissue tension etc is not provided. The authors should discuss this point. How many samples were assessed and are the obtained results statistically significant? Collagen alignment specifically appears very limited to assess and alternative methods as second harmonic imaging are more widely used to study collagen fibre orientation. The alignment measured in the illustrative pictures seems to be biased to regions with low density in which thick collagen bundles are visible but the finer network structure remains to be resolved. The authors should clarify and comment on this point and tone down conclusions accordingly.

Suppl.Fig2. Results from the viscoelastic model fitting should be included and discussed in the main manuscript.

The authors argue on several occasions about catch bond behaviour of actomyosin but direct evidence

is not provided and alternative mechanisms should be discussed as well.

Reviewer #2 (Remarks to the Author):

The manuscript is greatly improved, and all but one of my original concerns have been addressed. The outstanding concern is the data interpretation related to anisotropy, as explained more below:

1. Page 6: In the discussion of Figure 2, my concern about conflating the different types of anisotropy at play remains. The sentence, "This anisotropic contraction corresponds to a strongly anisotropic field of deformation, inferred from the displacement field," suggests that there are two forms of mechanical anisotropy that were measured: "contraction" and "deformation". However, the only contraction that was measured was a deformation field, so obviously the contraction and deformation (strain) correspond strongly to each other. It seems that much more is being made of the deformation results than what the measurements are able to provide.

2. Page 13: In the discussion of Figure 5, the interpretation of the data appears to be that anisotropy/isotropy of the deformation corresponds to the anisotropy/isotropy of the cytoskeletal and extracellular architecture. This interpretation comes from the observation that the architecture is more randomly oriented in the middle of the square but more preferentially aligned at the edge of the square, together with the observation that the strain field is more isotropic in the center than at the edge. However, without information on anisotropy in tissue stiffness in each of these regions, one does not know what is driving the anisotropy of the strain, since the strain results from both the contractility and the tissue stiffness. Moreover, in the four-pillar setup, the center region is likely in a different type of biaxial loading state than is the edge region, so I would expect the strain states could very well differ in these two regions even if the cytoskeletal and extracellular architecture were the same between regions.

Overall, I believe that the conclusions and suggestions in these two passages are a large over-reach and gloss over too many unexamined factors and likely alternative causes.

Reviewer #3 (Remarks to the Author):

The authors have addressed my concerns and I feel the manuscript is now ready for publication.

Response to the referees

Dear referees,

Thank you for your detailed and positive feedback on our recent submission to Nature Communications, ID NCOMMS-21-50274, " Light-driven biological actuators to probe the rheology of 3D microtissues". Although one of the reviewers feels the manuscript is ready for publication, two of you raised several good suggestions for improving the manuscript. We have now revised the manuscript accordingly, and feel that the changes address all of the concerns raised.

In particular, we have extensively modified the introduction and discussion sections to explicit the new insights obtained with our approach, and the abilities and limits of our technique, especially for characterizing tissue anisotropy. We also extensively extended the Materials and Methods section and we now provide a new supplemental figure and modified supplemental movies to facilitate the reproduction of our work and better illustrate the actin/collagen architecture in our microtissues.

Finally, we added new data in Fig. 1 to compare the tension and stress generated by microtissues composed of either opto-RhoA or wild type 3T3 fibroblasts and submitted to local light-stimulations.

Please find below a point-by-point response to the concerns you raised. All changes in the manuscript are highlighted in yellow.

Sincerely,
Thomas Boudou, Ph.D.

Reviewer #1: The authors have included several modifications in their manuscript and presented a detailed response to the raised questions. The paper keeps a technical focus and now includes validations of previously measured tissue parameters.

R1.1. New biological insight remains to be obtained with this method. As this is a major limitation of the current work the authors should consider to further strengthen in their manuscript which potential advantages their approach has over other previously used techniques.

Previously used techniques, based on external actuations, yielded key insights into how cells and ECM architecture impact microtissues' mechanics. However, using magnetic or vacuum actuations of cantilevers is restricted, by the fixed size of the cantilever, to the characterization of the global mechanical properties of microtissues, and rheological measurements have been shown to be very sensitive to the probe size and the loading history of the mechanical test device used for such characterization^{1,2}. Moreover, such externally applied forces are hardly comparable to cell-generated forces, which prevents their use to study the spatio-temporal regulation of cell forces in tissues. Biochemical treatments, on the other hand, allows the modulation of the signalization pathways responsible for these processes but their inability to target spatially-defined areas of tissues and their low temporal resolution severely limit their potential for further understanding how cell forces are generated, propagated and sensed in physiological and pathological tissues. Indeed, cells are constantly pulling and pushing on their microenvironment, thus assessing, among other things, its mechanical properties³. Probing the mechanical properties of the tissue "from the inside", as the cells naturally do it, would further our understanding of how cells investigate their environment and how cell-generated mechanical signals propagate.

By controlling locally, thanks to optogenetics, the generation of cell forces in engineered microtissues, our work shows how these mechanical signals propagate in fibrous tissues, and how they can be used to assess the rheology of 3D, physiological or pathological tissue models in real time and non-destructively, using their own constituting cells as internal actuators. We were thus able to demonstrate in fibrous tissue models (i) the dependence between local architecture, amplitude of cell-generated forces and the propagation of such forces, (ii) the roles of boundary conditions, collagen organization, tissue maturation and myofibroblastic differentiation on the mechanics of fibrous tissues, and (iii) the impact of cell forces on the formation and maturation of fibrous tissues.

Changes in the manuscript: We extensively modified the Introduction and Discussion sections of our manuscript (additions and changes highlighted in yellow) to explicit these new insights obtained with our approach and the advantages of our method over previously used techniques.

R1.2. Moreover, as the work by Mery et al. aims to establish a methodological advancement over previously established techniques to measure tissue mechanics the authors should further extend their methods section and technical description which remains very concise and mostly refers to previous work. This would make the work more transparent and reproducible for future applications which is not clearly doable at this stage.

We apologize our Materials and Methods section was not detailed enough. We understand from the reviewer's comment that our manuscript should provide more technical information and we have extended the Materials and Methods section to facilitate possible reproduction of our work.

Changes in the manuscript: We extensively modified the Materials and Methods section (additions and changes highlighted in yellow) and we now provide a new supplemental figure (Supp. Fig. 8) to illustrate it.

R1.3. Several points regarding the representation of data and conclusion drawn in the manuscript should also be considered by the authors as indicated below. Related to Fig1 and the assay

preparation, the authors revised panel 1A-B but should also include fluorescence microscopy images to show transversal/lateral views of microtissues and their attachment to cantilevers. It remains unclear from the method section how the contraction of tissues leads to their attachment to the cantilever tips. As the deflection of cantilevers is used to derive tissue tension, possible variations in tissue attachments to the cantilevers or friction with the bottom surface could alter measurements or induce high variability and the authors should clarify and clearly describe the methods in the manuscript. Detailed protocols on the methods including for example cell seeding density, quality criteria to assess proper microtissue formation and attachment etc should also be provided.

We now provide a supplemental figure showing quality criteria for selecting properly formed and anchored microtissues, top and side views of microtissues, and we modified the supplemental movie 3 to enhance its resolution and better illustrate the attachment of the tissue to the cantilevers. We also now describe in more details the seeding protocol, the formation of microtissues leading to their anchorage at the top of the cantilevers and the quality criteria to assess proper microtissue formation and attachment. Briefly, only tissues that were uniformly anchored to the tips of both cantilevers were included in the analysis. This anchorage between microtissue and cantilevers was visually checked over the whole duration of the experiment. Only microtissues wrapping completely the cap of both cantilevers were selected (Supp. Fig. 8). Tissues tearing or slipping from their cantilevers during an experiment were discarded from the analysis.

Changes in the manuscript: As stated before, we extensively modified the Materials and Methods section (additions and changes highlighted in yellow) and we now provide a new supplemental figure (Supp. Fig. 8) and a modified supplemental movie (Supp. Movie 3) to illustrate it.

R1.4. Were cantilever spring constants measured or calculated? How much was the cantilever deflection after 24h for wild type cells versus the 2% of cells selected for overexpressing the optogenetic construct to assess baseline tissue contraction in light vs dark conditions and the influence of the overexpression of the optogenetic construct.

We previously measured the spring constants of the cantilevers with a capacitive MEMS force sensor mounted on a micromanipulator^{4,5}. Briefly, the sensor tip was placed 20 μm below the top of the cap, the probe translated laterally against the outer edge of the cantilever, and the spring constant was calculated from the displacement of the cantilever head and the reported sensor force. We regularly assessed the cantilever geometry to ensure that it was not affected by successive replications. We now specify this point in the Materials and Methods section.

Due to overexpression of optoGEF-RhoA, microtissues composed of opto-RhoA fibroblasts exerted a higher baseline tension than microtissues composed of wild type (WT) 3T3 fibroblasts. However, opto-RhoA microtissues were also broader than WT microtissues and both microtissues generated similar baseline stress (Fig. 1.D-G). Of note, WT microtissues did not experience changes in tension upon illumination, similarly to previous work with single cells⁶. We modified Fig. 1 comparing the tension and stress generated by microtissues composed of either Opto-Rhoa or WT 3T3 fibroblasts, and submitted to left and right stimulations.

Changes in the manuscript: We added a description of the cantilever calibration in the Materials and Methods section and we modified Fig. 1 comparing the tension and stress generated by microtissues composed of either Opto-Rhoa or WT 3T3 fibroblasts. We also added the following sentence p.4:

" By engineering microtissues composed of wild type (WT) NIH-3T3 fibroblasts, we confirmed these light induced contractions were due to the optogenetic construction and inexistent in WT microtissues (Fig. 1.D-F). Of note, opto-Rhoa microtissues were broader and generated more tension than WT microtissues, but both generated similar baseline stress σ_{xx} (i.e. tissue tension obtained from the pillar deflection divided by the cross-sectional area in the center of the tissue) (Fig. 1.G). "

R1.5. Fig 1E, the authors should connect tension measured in this panel vs Fig 3C. The stress derivation based on area measures should be exemplified. How and when is area change measured? And how is the area measure obtained in asymmetric vs central stimulation when the optogenetic activation of contraction can be expected to lead to changes in area?

We understand from the reviewer's comment that the stress derivation was not explained clearly enough. The stress σ_{xx} was calculated by dividing the force F , obtained from the deflection d of the cantilevers and their spring constant k ($F = k \cdot d$), by the cross-sectional area measured in the center of the tissue. The width of each microtissue (i.e. its dimension along the y-axis, the x-axis being between the two cantilevers) was tracked during stimulation experiments. The cross-sectional areas were measured from confocal z-stack images of tissues fixed and stained right after experiments, in their center, similarly to previous works (Supp. Fig. 8) ^{4,7-11}. No measurable change in cross-sectional area was observed over the duration of light-stimulation experiments, in agreement with the almost inexistent displacements along the y-axis (Fig. 1-4). Consequently, the cross-sectional areas were considered unchanged over the duration of stimulation. The cross-sectional areas were linearly correlated with the squared tissue width w^2 (Supp. Fig. 8.H, $R^2 = 0.80$), which allowed to estimate tissue cross-section when microtissues could not be fixed immediately after experiments. We also added the stress evolution over time for both opto-RhoA and WT microtissues in Fig. 1. Of note, the baseline tension of the opto-RhoA is higher in the new Fig. 1 compared to the previous one as we used stiffer cantilevers, with the same spring constant than in Fig. 3 to connect both figures.

Changes in the manuscript: We extended the description of the stress derivation in the Materials and Methods section, we added a new supplemental figure (Supp. Fig. 8) to support it, and we added the stress quantification of opto-RhoA and WT microtissues stimulated with left and right pulses in Fig. 1.

R1.6. Fig 2G-L and Fig5L-P, the imaging quality and resolution to assess actin and collagen alignment remain limited. Also cell density appears more variable in these panels as for example in Suppl.Fig1 and it appears cells are rounder in the middle region of the tissue versus the margin but a spatial map and how it links to tissue tension etc is not provided. The authors should discuss this point. How many samples were assessed and are the obtained results statistically significant? Collagen alignment specifically appears very limited to assess and alternative methods as second harmonic imaging are more widely used to study collagen fibre orientation. The alignment measured in the illustrative pictures seems to be biased to regions with low density in which thick collagen bundles are visible but the finer network structure remains to be resolved. The authors should clarify and comment on this point and tone down conclusions accordingly.

The numbers of samples are stated in the figure captions, 24 samples for Fig. 2.G-L and 16 samples for Fig. 5.L-P. The alignments of actin and collagen are significantly different between the center and the side of the microtissues in Fig. 5.L-P, we added the significance information in the caption of the figure. Cell density appears more variable in Fig. 2.G-L and Fig. 5.L-P because, as stated in the captions, these are single confocal slices whereas Supp. Fig. 1 presents confocal projections (i.e. average of z-stacks). Cells are indeed slightly rounder and sparser in the core of the tissue compared to cells on the periphery (see. Supp. Movie 3), as previously observed ^{5,9}. We now discuss this point in the manuscript. Imaging simultaneously thick collagen bundles and fine, single collagen fibers in 100 μ m-thick microtissues is indeed complex, as is representing such data (e.g. fine collagen fibers cannot be visualized without saturating the fluorescence of the thick collagen bundles). Second harmonic microscopy is an interesting suggestion for future work as it could yield a more detailed quantification of the collagen organization in microtissues, and we now state this point in the discussion section. However, confocal imaging of immuno-stained collagen already evidences very clear tendencies in collagen alignment and drastic differences between aligned and randomly oriented collagen fibers in square microtissues. Although second harmonic would allow a more accurate quantification, it would not change the conclusions drawn from the results presented in Fig. 2.G-L and Fig. 5.L-P. Previous works used similar confocal imaging of immuno-stained collagen to thoroughly characterize the

structural aspect of collagen in microtissues composed of 3T3 fibroblasts embedded in collagen and obtained similar results to ours, in 2 and 4 cantilever microtissue ^{5,11}. Nevertheless, we now provide higher quality Supp. Movie 3 and 8 that better illustrate the resolution of our confocal imaging. Moreover, we added in Supp. Movie 8 magnified stacks of the left and center regions of a representative square microtissue, with a gamma correction that allows better visualization of thin dim fibers without saturating the thick bundles. Finally, we now extensively detail in the Materials and Methods section how actin and collagen organization were quantified with the Orientation J plugin (<http://bigwww.epfl.ch/demo/orientationj/>) ¹² in Image J. This plugin computes the structure tensor for each pixel in the image by sliding a Gaussian analysis window (variance $\sigma = 2$ pixels) over the entire image. From the structure tensor are extracted both the orientation and coherency properties of the region of interest. These properties are then gathered in a color map in HSB (Hue Saturation Brightness) mode where the hue corresponds to the orientation, the saturation to the coherency and the brightness to the source image. The coherency indicates if the local image features are coherently oriented or not. The histograms presented in the manuscript are weighted histograms, the weight being the coherency. Consequently, thicker or denser collagen bundles are weighted more than isolated, randomly oriented fibers, in coherence with the fact that thick and dense collagen bundles weight more in the mechanical behavior of a fibrous microtissue than thin random fibers ¹¹.

Changes in the manuscript: We now detail how actin and collagen organization were quantified with the Orientation J plugin (highlighted in yellow in the Materials and Methods section) and we added the following sentences:

p.6: " The confocal images showed a compacted collagen core, sparsely populated with fibroblasts, surrounded by a highly cellularized peripheral shell (Supp. movie 3), consistent with previous observations ^{5,9}."

p. 16: "Data of actin and collagen orientations are presented as mean \pm SD with n = 16 microtissues and are significantly different (****p < 0.0001)."

p. 18: " As collagen organization has been shown to be key to the long-scale interactions between cells and their navigation through the ECM ¹³⁻¹⁵, a more precise characterization using second-harmonic microscopy would be key to delineate the respective roles of thin isolated fibers and thick, strongly anisotropic collagen bundles. "

R1.7. Suppl.Fig2. Results from the viscoelastic model fitting should be included and discussed in the main manuscript.

We included and discussed the results from the viscoelastic model fitting in the manuscript.

Changes in the manuscript: We added the following sentences:

p. 10: " Fitting such models to the data presented in Figure 4 for a stimulation width of 50 μm lead to consistent relaxation times τ between 101 and 135 s for the three different models. Although the Kelvin Voigt model captured the overall shape of the mechanical response and gave an elastic modulus of 18.7 kPa close to our experimental measurements, it failed to capture both the short- and long-time behavior. The SLS and the SE models lead to similar elastic constants ($E_1 = 56.3$ kPa and $E_2 = 14.7$ kPa for the SLS model, $E_1 = 65.0$ kPa and $E_2 = 10.0$ kPa for the SE model), but the SE model allowed for better fitting the long-time recovery response, thanks to the dimensionless constant $\beta = 0.66$ that captures a specific distribution of timescales (i.e. when $\beta = 1$, the SE model behaves as a SLS model, whereas when β decreases, the distribution of timescales broadens). Such distribution is similar to previously obtained results ¹⁶ and describes the broad distribution of inter-related timescales inherent to the viscoelastic heterogeneities of the different microtissue components."

R1.8. The authors argue on several occasions about catch bond behaviour of actomyosin but direct evidence is not provided and alternative mechanisms should be discussed as well.

We now list different hypotheses for the stress-dependent dynamic of strain propagation in fibrous tissues and discuss their plausibility in view of our results and other findings from the literature.

Changes in the manuscript:

p. 17: " These results thus evidence a stress-dependent dynamic of strain propagation in fibrous microtissues, similarly to previous work in fibroblast populated collagen matrices that described an increase of the loss modulus with the strain rate ¹⁷ or an increase in stress-strain hysteresis with the strain rate ¹⁸, or in presence of myofibroblasts ¹⁹. One hypothesis for this behavior is a dependence between amplitude and dynamics of stress. However, we did not measure any difference in the time from zero to maximum contraction stress when varying the light intensity or the cantilever spring constant. A second hypothesis is an inherent stress-dependent viscosity of the collagen matrix but the inexistent difference in τ_s when the collagen density is varied or when the collagen is cleaved with collagenase proves otherwise. A third hypothesis is the catch bond behavior of the contractile cytoskeleton that may induce such stress-dependent strain propagation. Although the catch-bond behavior of actin-myosin or cadherin-catenin bonds was demonstrated in single molecules ^{20,21} or single cells experiments ²², our results motivate further experimental studies to test whether this hypothesis is indeed correct in fibrous tissues, possibly via the use of cytoskeleton-associated tension reporters ²³⁻²⁵."

Reviewer #2:

The manuscript is greatly improved, and all but one of my original concerns have been addressed. The outstanding concern is the data interpretation related to anisotropy, as explained more below:

R2.1. Page 6: In the discussion of Figure 2, my concern about conflating the different types of anisotropy at play remains. The sentence, "This anisotropic contraction corresponds to a strongly anisotropic field of deformation, inferred from the displacement field," suggests that there are two forms of mechanical anisotropy that were measured: "contraction" and "deformation". However, the only contraction that was measured was a deformation field, so obviously the contraction and deformation (strain) correspond strongly to each other. It seems that much more is being made of the deformation results than what the measurements are able to provide. We understand from the reviewer's comment that the formulation we used was misleading. Indeed, our setup only gives access to the anisotropy of the deformation field and, as we stated in our previous response, we cannot delineate the different forms of anisotropy (anisotropy of cell contractions, anisotropy of cellular mechanical properties, and anisotropy of fiber orientations). To our knowledge, there is currently no experimental setup allowing such delineation. In the revised manuscript, we have clarified the limits of our approach for assessing anisotropy and discussed possible ways to address such complex mechanical characterization in the discussion section (see also our answer to the next comment).

Changes in the manuscript: We modified the following sentences:

p. 6: "Despite a discoidal light stimulation, the resulting displacements were strongly polarized, with a mean angle of $16 \pm 7^\circ$ and more than 80% of the displacements presenting a lower than 30° angle with the longitudinal x-axis of the tissue (Fig. 2.C). This polarized displacement field corresponds to a strongly anisotropic field of deformation, as the x-component of the strain (ϵ_{xx}) was more than 15-times larger than its y-component (ϵ_{yy})"

R2.2. Page 13: In the discussion of Figure 5, the interpretation of the data appears to be that anisotropy/isotropy of the deformation corresponds to the anisotropy/isotropy of the cytoskeletal and extracellular architecture. This interpretation comes from the observation that the architecture is more randomly oriented in the middle of the square but more preferentially aligned at the edge of the square, together with the observation that the strain field is more isotropic in the center than at the edge. However, without information on anisotropy in tissue stiffness in each of these regions, one does not know what is driving the anisotropy of the strain, since the strain results from both the contractility and the tissue stiffness. Moreover, in the four-pillar setup, the center region is likely in a different type of biaxial loading state than is the edge region, so I would expect the strain states could very well differ in these two regions even if the cytoskeletal and extracellular architecture were the same between regions. Overall, I believe that the conclusions and suggestions in these two passages are a large over-reach and gloss over too many unexamined factors and likely alternative causes.

We agree with the reviewer that the experiments we showed do not allow to determine the cause of the anisotropy, whether it is due to the cytoskeletal and extracellular organization or the spatial heterogeneity in stiffness, which is why we only stated that cellular/extracellular organization correlated with the strain anisotropy. The strain pattern we observed is very possibly the consequence of a combination of both parameters, as previous work showed that microtissue regions with aligned actin and collagen fibers were significantly stiffer than randomly organized regions¹¹. However, delineating the different types of anisotropy remains currently, to our knowledge, an extremely complex task that has only begun to be undertaken, and would require further experimental studies,

possibly combining tissue engineering, optogenetics, cytoskeleton-associated tension reporters and computational modeling^{5,11,23–25}. We now discuss this point in more details in the revised manuscript.

Changes in the manuscript: We added the following sentences:

p. 15: "Bose et al. previously demonstrated that both ECM fiber alignment and density, resulting from the history of tissue formation, influence the local tissue stiffness¹¹. The correlation between the strain pattern, probed via optogenetics, the fine-scale cytoskeletal and extracellular architecture, and possibly stiffness heterogeneities thus demonstrates that the spatial propagation of mechanical signals in microtissues is strongly dependent upon actin and collagen organization, which in turn depend on the formation history of the tissue."

p.19: "Moreover, although we observed a correlation between cell-induced strain patterns and tissue architecture, we cannot deduce from our experiment the cause of the anisotropic strain pattern, whether it is due to the cytoskeletal and extracellular organization, the spatial heterogeneity in stiffness or a combination of both parameters. Microtissue regions with aligned actin and collagen fibers were shown to be significantly stiffer than randomly organized regions¹¹, but unraveling the respective roles of the different types of anisotropy (i.e. anisotropy of cellular contraction, of mechanical properties, of fiber orientations) remains currently an extremely complex task that has only begun to be undertaken, and would require further experimental studies, possibly combining tissue engineering, optogenetics, cytoskeleton-associated tension reporters and computational modeling^{5,11,23–25}."

References

1. Wu, P.-H. *et al.* A comparison of methods to assess cell mechanical properties. *Nat. Methods* **1** (2018). doi:10.1038/s41592-018-0015-1
2. Licup, A. J. *et al.* Stress controls the mechanics of collagen networks. *Proc. Natl. Acad. Sci.* **112**, 9573–9578 (2015).
3. Eyckmans, J., Boudou, T., Yu, X. & Chen, C. S. A hitchhiker's guide to mechanobiology. *Dev. Biol.* **21**, 35–47 (2011).
4. Boudou, T. *et al.* A Microfabricated Platform to Measure and Manipulate the Mechanics of Engineered Cardiac Microtissues. *Tissue Eng. Part A* **18**, 910–919 (2012).
5. Legant, W. R., Chen, C. S. & Vogel, V. Force-induced fibronectin assembly and matrix remodeling in a 3D microtissue model of tissue morphogenesis. *Integr. Biol.* **4**, 1164 (2012).
6. Valon, L., Marín-Llauradó, A., Wyatt, T., Charras, G. & Trepap, X. Optogenetic control of cellular forces and mechanotransduction. *Nat. Commun.* **8**, 14396 (2017).
7. Legant, W. R. *et al.* Microfabricated tissue gauges to measure and manipulate forces from 3D microtissues. *Proc. Natl. Acad. Sci. USA* **106**, 10097–10102 (2009).
8. Zhao, R., Boudou, T., Wang, W.-G., Chen, C. S. & Reich, D. H. Decoupling Cell and Matrix Mechanics in Engineered Microtissues Using Magnetically Actuated Microcantilevers. *Adv. Mater.* **25**, 1699–1705 (2013).
9. Zhao, R., Chen, C. S. & Reich, D. H. Force-driven evolution of mesoscale structure in engineered 3D microtissues and the modulation of tissue stiffening. *Biomaterials* **35**, 5056–5064 (2014).
10. Asmani, M. *et al.* Fibrotic microtissue array to predict anti-fibrosis drug efficacy. *Nat. Commun.* **9**, 1–12 (2018).
11. Bose, P., Eyckmans, J., Nguyen, T. D., Chen, C. S. & Reich, D. H. Effects of Geometry on the Mechanics and Alignment of Three-Dimensional Engineered Microtissues. *ACS Biomater. Sci. Eng.* **5**, 3843–3855 (2019).
12. Rezakhaniha, R. *et al.* Experimental investigation of collagen waviness and orientation in the arterial adventitia using confocal laser scanning microscopy. *Biomech. Model. Mechanobiol.* **11**, 461–473 (2012).
13. Gjorevski, N., S. Piotrowski, A., Varner, V. D. & Nelson, C. M. Dynamic tensile forces drive

- collective cell migration through three-dimensional extracellular matrices. *Sci. Rep.* **5**, 11458 (2015).
14. Pakshir, P. *et al.* Dynamic fibroblast contractions attract remote macrophages in fibrillar collagen matrix. *Nat. Commun.* **10**, 1850 (2019).
 15. Ahmadzadeh, H. *et al.* Modeling the two-way feedback between contractility and matrix realignment reveals a nonlinear mode of cancer cell invasion. *Proc. Natl. Acad. Sci.* **114**, E1617–E1626 (2017).
 16. Walker, M., Godin, M., Harden, J. L. & Pelling, A. E. Time dependent stress relaxation and recovery in mechanically strained 3D microtissues. *APL Bioeng.* **4**, 036107 (2020).
 17. Wakatsuki, T., Kolodney, M. S., Zahalak, G. I. & Elson, E. L. Cell mechanics studied by a reconstituted model tissue. *Biophys. J.* **79**, 2353–2368 (2000).
 18. Wagenseil, J. E., Wakatsuki, T., Okamoto, R. J., Zahalak, G. I. & Elson, E. L. One-Dimensional Viscoelastic Behavior of Fibroblast Populated Collagen Matrices. *J. Biomech. Eng.* **125**, 719–725 (2003).
 19. Walker, M., Godin, M. & Pelling, A. E. Mechanical stretch sustains myofibroblast phenotype and function in microtissues through latent TGF- β 1 activation. *Integr. Biol. (Camb)*. **12**, 199–210 (2020).
 20. Buckley, C. D. *et al.* The minimal cadherin-catenin complex binds to actin filaments under force. *Science (80-.)*. **346**, (2014).
 21. Guo, B. & Guilford, W. H. Mechanics of actomyosin bonds in different nucleotide states are tuned to muscle contraction. *Proc. Natl. Acad. Sci. U. S. A.* **103**, 9844–9849 (2006).
 22. Vernerey, F. J. & Akalp, U. Role of catch bonds in actomyosin mechanics and cell mechanosensitivity. *Phys. Rev. E* **94**, 012403 (2016).
 23. Borghi, N. *et al.* E-cadherin is under constitutive actomyosin-generated tension that is increased at cell-cell contacts upon externally applied stretch. *Proc. Natl. Acad. Sci. USA* **109**, 12568–12573 (2012).
 24. Ringer, P. *et al.* Multiplexing molecular tension sensors reveals piconewton force gradient across talin-1. *Nat. Methods* **14**, (2017).
 25. Colom, A. *et al.* A fluorescent membrane tension probe. *Nat. Chem.* **10**, 1118–1125 (2018).

REVIEWERS' COMMENTS

Reviewer #1 (Remarks to the Author):

The authors have now sufficiently addressed all my concerns. The revised manuscript now includes a detailed method description, additional data and extended discussion sections that improved the presentation and readability of the work and the manuscript is from my perspective ready for publication.

ON REVIEWER #2:

Reviewer #2 commented on two different types of anisotropy possibly originating from cellular versus matrix characteristics that can impact on the mechanical response of tissues.

In my opinion the authors correctly argue that the main readout of their measurements is the deformation field and their data allow to establish a correlation between the measured strain anisotropy and cell/matrix organization which are assessed by imaging methods and perturbation assays.

The authors revised their arguments in the main manuscript and now clearly state that different factors can cause the measured anisotropy of the deformation field, including cytoskeletal and extracellular matrix organization as well as heterogeneities in cell contraction (p.15, p.19). This makes the reader aware of different sources that influence the mechanical response of tissue. Their work also motivates future studies how extracellular versus cellular properties and their intricate coupling control the global response of tissues to mechanical forces.

Response to the referees

Dear referees,

Thank you for your positive feedback on our recent submission to Nature Communications, ID NCOMMS-21-50274, " Light-driven biological actuators to probe the rheology of 3D microtissues":

"Reviewer #1: The authors have now sufficiently addressed all my concerns. The revised manuscript now includes a detailed method description, additional data and extended discussion sections that improved the presentation and readability of the work and the manuscript is from my perspective ready for publication."

"On Reviewer #2: Reviewer #2 commented on two different types of anisotropy possibly originating from cellular versus matrix characteristics that can impact on the mechanical response of tissues. In my opinion the authors correctly argue that the main readout of their measurements is the deformation field and their data allow to establish a correlation between the measured strain anisotropy and cell/matrix organization which are assessed by imaging methods and perturbation assays. The authors revised their arguments in the main manuscript and now clearly state that different factors can cause the measured anisotropy of the deformation field, including cytoskeletal and extracellular matrix organization as well as heterogeneities in cell contraction (p.15, p.19). This makes the reader aware of different sources that influence the mechanical response of tissue. Their work also motivates future studies how extracellular versus cellular properties and their intricate coupling control the global response of tissues to mechanical forces."

We are delighted by your comments and the resulting decision of the editor to accept our work for publication. Thank you again for your insightful comments throughout the revision, they have greatly helped us improve our manuscript.

Sincerely,
Thomas Boudou, Ph.D.